# Enhanced thermally-activated skyrmion diffusion with tunable effective gyrotropic force

Takaaki Dohi [1,2] ✉, Markus Weißenhofer [3,4,5] ✉, Nico Kerber [1,6],
Fabian Kammerbauer [1], Yuqing Ge[1], Klaus Raab [1], Jakub Zázvorka[7],
Maria-Andromachi Syskaki[1,8], Aga Shahee[1], Moritz Ruhwedel[9], Tobias Böttcher[6,9],
Philipp Pirro [9], Gerhard Jakob [1,6], Ulrich Nowak [3] & Mathias Kläui[1,6] ✉

Magnetic skyrmions, topologically-stabilized spin textures that emerge in magnetic systems, have garnered considerable interest due to a variety of electromagnetic responses that are governed by the topology. The topology that creates a microscopic gyrotropic force also causes detrimental effects, such as the skyrmion Hall effect, which is a well-studied phenomenon highlighting the influence of topology on the deterministic dynamics and drift motion. Furthermore, the gyrotropic force is anticipated to have a substantial impact on stochastic diffusive motion; however, the predicted repercussions have yet to be demonstrated, even qualitatively. Here we demonstrate enhanced thermally-activated diffusive motion of skyrmions in a specifically designed synthetic antiferromagnet. Suppressing the effective gyrotropic force by tuning the angular momentum compensation leads to a more than 10 times enhanced diffusion coefficient compared to that of ferromagnetic skyrmions. Consequently, our findings not only demonstrate the gyro-force dependence of the diffusion coefficient but also enable ultimately energy-efficient unconventional stochastic computing.

Magnetic skyrmions are topologically-stabilized spin textures[1–8] that exhibit intriguing dynamics governed by their topology[9–13]. In particular, in thin film systems, it has been well known that magnetic skyrmions can be stabilized above room temperature with fixed chirality determined by the Dzyaloshinskii–Moriya interaction (DMI) induced at the heavy metal (HM)/ferromagnet (FM) interface[4,14–18]. Simultaneously, the adjacent HM produces a spin-orbit torque (SOT)[19], allowing for efficient manipulation of the magnetic skyrmions by electrical means, which is an essential function for skyrmion electronics often termed skyrmionics[4,5,8,20,21].

Recent material development of ultralow pinning systems hosting magnetic skyrmion has led to the successful observation of thermally-activated diffusive skyrmion dynamics[22–25]. The magnetic skyrmions exhibit Brownian-like motion driven by thermal fluctuations, leading to a linear time dependence of the mean-squared displacement (MSD)[22,26–29]. Such stochastic dynamics of the magnetic skyrmions have been suggested for ultimately energy-efficient unconventional computing[22,30,31].

So far, most studies of the impact of skyrmion topology on the dynamics have focused on deterministic current-driven skyrmion

[1]Institut für Physik, Johannes Gutenberg-Universität Mainz, Staudingerweg 7, 55128 Mainz, Germany. [2]Laboratory for Nanoelectronics and Spintronics, Research Institute of Electrical Communication, Tohoku University, Sendai 980–8577, Japan. [3]Fachbereich Physik, Universität Konstanz, DE-78457 Konstanz, Germany. [4]Department of Physics and Astronomy, Uppsala University, P.O. Box 516, S-751 20 Uppsala, Sweden. [5]Department of Physics, Freie Universität Berlin, Arnimallee 14, D-14195 Berlin, Germany. [6]Graduate School of Excellence Materials Science in Mainz, Staudingerweg 9, 55128 Mainz, Germany. [7]Institute of Physics, Faculty of Mathematics and Physics, Charles University, Ke Karlovu 5, Prague 12116, Czech Republic. [8]Singulus Technologies AG, 63796 Kahl am Main, Germany. [9]Fachbereich Physik und Landesforschungszentrum OPTIMAS, Technische Universität Kaiserslautern, Gottlieb-Daimler-Straße 46, 67663 Kaiserslautern, Germany. ✉e-mail: tdohi@tohoku.ac.jp; markus.weissenhofer@uni-konstanz.de; klaeui@uni-mainz.de

motion. For such drift motion, the skyrmion Hall effect[11,12,32–34], which is a perpendicular motion component to the current flow direction, has been observed. This is an archetypal example of the strong topology dependence of magnetic skyrmion dynamics. While the effect of topology on the deterministic current-induced drift motion of magnetic skyrmions has been well studied, for the diffusive motion regime, so far, only intriguing theoretical predictions have been made. In particular, it was calculated that the microscopic gyrotropic force originating from the finite topology gives rise to a drastic decrease of the diffusion coefficient via a different dependence on the damping as compared to topologically trivial structures[26–28]. Hence, one needs to be able to tailor the effective gyrotropic force to maximize diffusive dynamics.

A possible approach is to use antiferromagnetically-coupled skyrmions. While the skyrmion topology is defined by the Néel vector, the compensation of angular momentum in the two sub-lattices allows for the control of the effective gyrotropic force generated by the topology[35–41]. However, typical intrinsic crystalline antiferromagnets or ferrimagnets have been found to exhibit strong pinning[40,41], making these systems unsuitable for the observation of diffusive motion.

Here we demonstrate that an amorphous-like synthetic antiferromagnetic (SyAFM) system with low pinning enables us to observe a thermally-activated diffusive motion of antiferromagnetically-coupled skyrmions. The systematic investigation, varying the compensation ratio of magnetic moments in the magnetic layers, allows us to tune the microscopic gyrotropic force flexibly. By accounting for pinning effects, we can directly demonstrate the influence of compensation on the diffusive motion. Our analysis reveals a more than 10 times larger diffusion coefficient for highly compensated antiferromagnetically-coupled skyrmions, which is a direct consequence of the reduction of the effective gyrotropic force stemming from the topological charge, which provides crucial insights into the thermally-activated dynamics of the topological objects in antiferromagnetic systems.

## Results

### Experimental setup and magnetic properties

A schematic of the experimental setup is shown in Fig. 1a, where magnetic thin films are placed on Peltier modules to investigate the diffusive motion of magnetic skyrmions imaged by a magneto-optical Kerr effect (MOKE) microscope in a polar configuration. While ferrimagnets or crystalline intrinsic antiferromagnets have been shown to exhibit strong skyrmion pinning[40,41], the advantage of SyAFM systems is to use low pinning magnetic materials such as CoFeB[42]. Our SyAFM systems consist of $Ta(5.00)/Pt(1.03)/Co_{0.60}Fe_{0.20}B_{0.20}(t_{CFB1})/Co_{0.20}Fe_{0.60}B_{0.20}(t_{FCB1})/Ir(1.20)/Co_{0.60}Fe_{0.20}B_{0.20}(t_{CFB2})/Co_{0.20}Fe_{0.60}B_{0.20}(t_{FCB2})/Ru(1.00)$ (in nm) where $t_{CFB}$ and $t_{FCB}$ are tuned to control magnetic properties such as the compensation ratio of magnetic moments (see "Methods" for more details). As a reference, a FM bi-layer stack consisting of $Ta(5.00)/Pt(1.03)/Co_{0.60}Fe_{0.20}B_{0.20}(0.50)/$

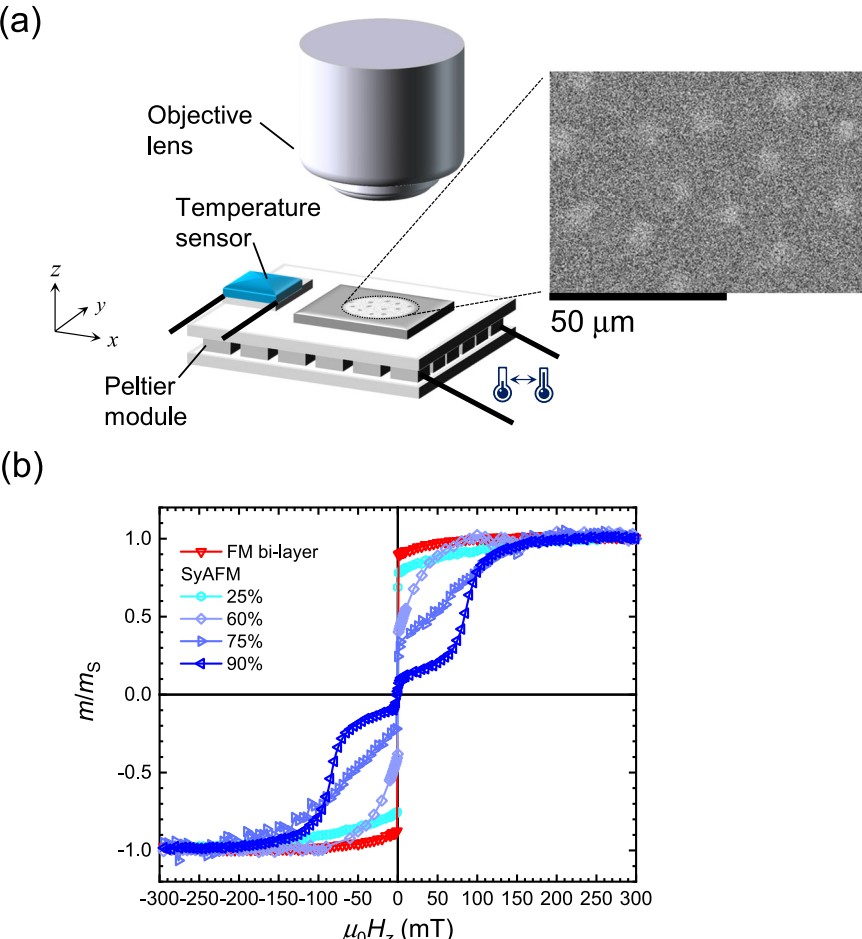

**Fig. 1 | Experimental setup and measurements. a** A schematic of the experimental setup. Magnetic thin films are placed on a Peltier module. The magnetic domain is observed using a magneto-optical Kerr-effect microscope in a polar configuration. The magnified picture shows the observed SyAFM skyrmions ($m_{Com} = 75\%$) at 320.7 K under an applied field of $\mu_0 H_z = 0.35$ mT. **b** The $m–H_z$ curves for ferromagnetic bi-layer and synthetic antiferromagnetic systems with various compensation ratios at room temperature. The red color and the blue colors correspond to FM bi-layer and the SyAFM systems with 25% (very light blue), 60% (light blue), 75% (intermediate blue), and 90% (dark blue) compensation ratios, respectively.

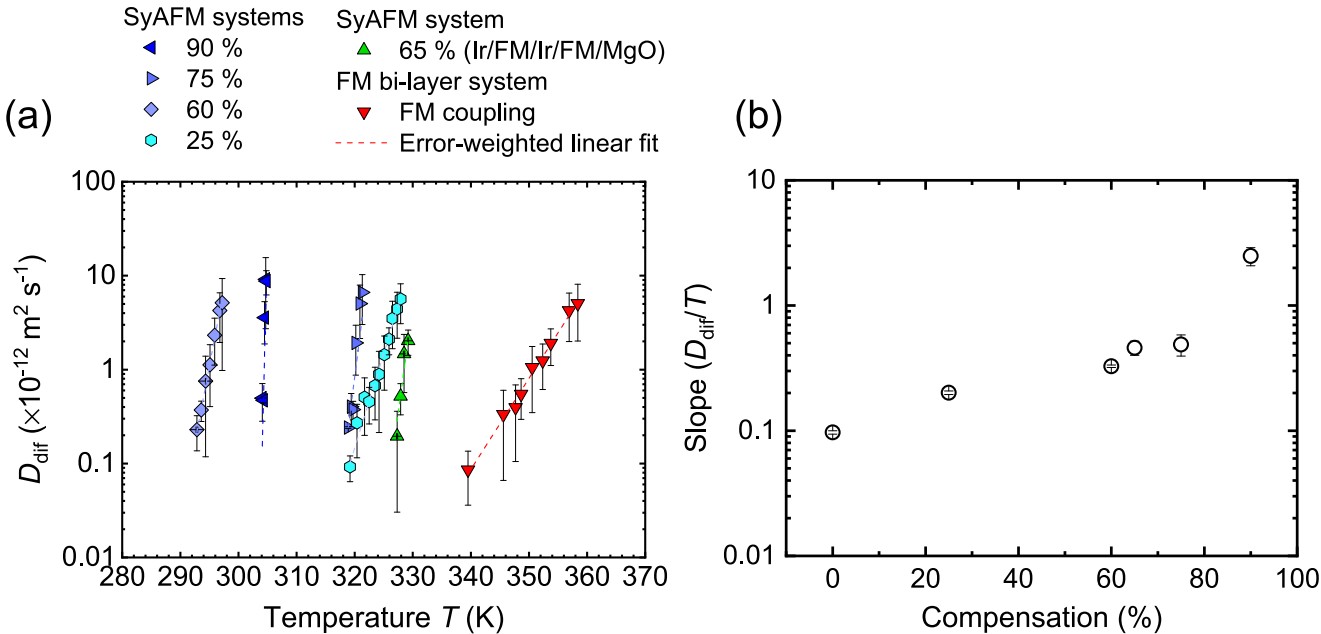

**Fig. 2 | Thermally-activated diffusive motion of SyAFM skyrmions. a** The temperature dependence of the diffusion coefficient for the FM bi-layer and the SyAFM systems with various compensation ratios. Red and the other colors represent the FM bi-layer and the SyAFM systems, respectively. The green plot indicates a different type of material system, $Ir/Co_{0.40}Fe_{0.40}B_{0.20}/Ir/Co_{0.20}Fe_{0.60}B_{0.20}/MgO/Ta$. The dashed line denotes an error-weighted linear fit. **b** The compensation ratio dependence of the slope ($D_{dif}/T$) for Fig. 2a.

$Co_{0.20}Fe_{0.60}B_{0.20}(0.35)/Ir(1.60)/Co_{0.60}Fe_{0.20}B_{0.20}(0.50)/Co_{0.20}Fe_{0.60}B_{0.20}(0.35)/Ru(1.00)$ is prepared. The Ir thickness is controlled to tailor the sign and the strength of interlayer exchange coupling (see Supplementary Note 1).

Figure 1b shows magnetization curves ($m-H$ curves) for the FM bi-layer and SyAFM systems, where the compensation ratio $m_{Com} = 1 - |m_1 + m_2|/(|m_1| + |m_2|)$, the saturation magnetic moment of the bottom FM layer $m_1$, and that of the top FM layer $m_2$ can be determined[37]. We intentionally tailor the maximum compensation ratio to be ~90% to still allow for the observation using MOKE while keeping $m_2 > m_1$ to make use of the surface sensitivity[43] (see "Methods", and Supplementary Note 2). As shown in Fig. 1b, all used SyAFM systems exhibit a spin-flop-like transition indicating that the effective magnetic anisotropy energy density is smaller than the interlayer exchange coupling $K_{eff} < -J_{int}/(t_{CFB} + t_{FCB})$, facilitating the emergence of antiferromagnetically-coupled magnetic domain states[37,38,44], which however stay always coupled in our experiments (see Supplementary Note 3).

### Thermally-activated skyrmion diffusion and its temperature dependence

Magnetic skyrmions are nucleated using magnetic in-plane field pulses, where the strength of the pulses is tuned to control the density of magnetic skyrmions. The magnified picture in Fig. 1a shows the observed SyAFM skyrmions for $m_{Com} = 75\%$ at $T = 320.7$ K. We find that the SyAFM skyrmions clearly exhibit thermally-activated diffusive motion (see Supplementary Movie 1). The SyAFM skyrmions are tracked similarly as established for FM skyrmions[22] (see "Methods"). We confirm that the mean-squared displacement (MSD) of individual skyrmions represents the linear behavior as a function of time $t$, indicating diffusive Brownian-like motion. The diffusion coefficient $D_{dif}$ for the Brownian motion in two-dimensional ($xy$) coordinate systems reads the MSD $= <x^2(t) + y^2(t)> = 4D_{dif}t$. We determine the diffusion coefficient from the slope of the MSD at a specific temperature, which is also a well-established method for characterizing $D_{dif}$[22–25,31]. Figure 2a shows the temperature dependence of the diffusion coefficients for each stack. Note that the measurable temperature range is limited due to constraints regarding the skyrmion nucleation, stability, and the time resolution of our imaging setup. We find that the diffusion coefficient for all the stacks exhibits a linear dependence on a semi-logarithmic scale, which is analogous to the ferromagnetic counterpart in a single layer[22–24]. Moreover, we reveal that the slope becomes steeper with increasing $m_{Com}$ as seen in Fig. 2a, b. The highest compensation ($m_{Com} = 90\%$) clearly exhibits the steepest slope in the temperature dependence. Intriguingly, a different material system with a compensation ratio of 65%, consisting of $Ta(5.00)/Ir(1.20)/Co_{0.40}Fe_{0.40}B_{0.20}(1.00)/Ir(1.20)/Co_{0.20}Fe_{0.60}B_{0.20}(0.80)/Ta(0.09)/MgO(2.00)/Ta(5.00)$, shown in light green in Fig. 2a, exhibits similar behavior to that of $m_{Com}$ (60 or 75%). This implies that the steeper slope with the compensation is not a material-specific behavior but rather a more general feature resulting from the compensation. To clarify the intrinsic effect of the compensation on the diffusive motion and to understand the intriguing temperature dependence, we next perform atomistic spin simulations and establish an analytical description of both the compensation dependence as well as pinning effects.

### Theoretical calculations

Atomistic spin simulations[45] within a toy model system (for details, see "Methods") are performed first in the absence of pinning. We consider two ferromagnetic monolayers that are either ferro-, or antiferromagnetically coupled. The compensation of the synthetic antiferromagnet is varied by keeping magnetization (the magnetic moments per volume) $M_2$ fixed and varying $M_1$ between $-M_2$ and 0. The ferromagnetic bilayer is described by $M_1 = M_2$. Note that the skyrmions described by the toy model are small compared to the ones in the experiment. This, however, is not a problem because the atomistic spin simulations are merely used to demonstrate the validity of an analytic theory that holds irrespective of the skyrmion size as discussed next.

Figure 3a displays the diffusion coefficient of the SyAFM skyrmions with $M_2 = -M_1$ ($m_{Com} = 100\%$), $M_2 = -0.75M_1$ ($m_{Com} = 85.7\%$), $M_2 = -0.5M_1$ ($m_{Com} = 66.6\%$), $M_2 = -0.25M_1$ ($m_{Com} = 40\%$) and the FM skyrmions versus temperature in a semi-logarithmic plot. This data can be well described by the following formula which is derived from an

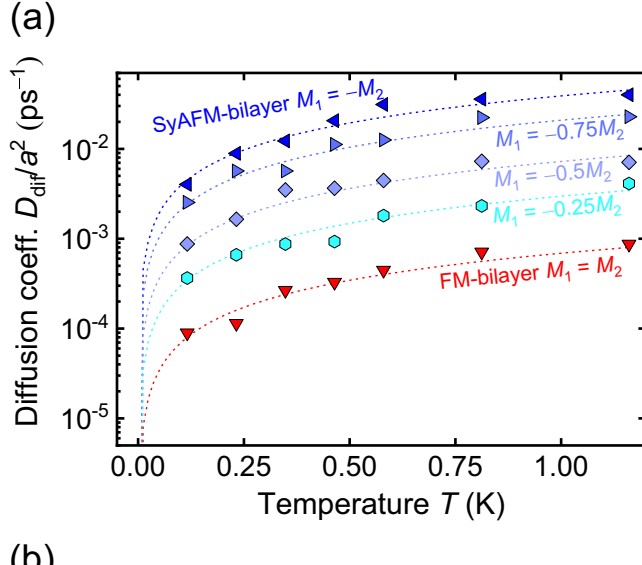

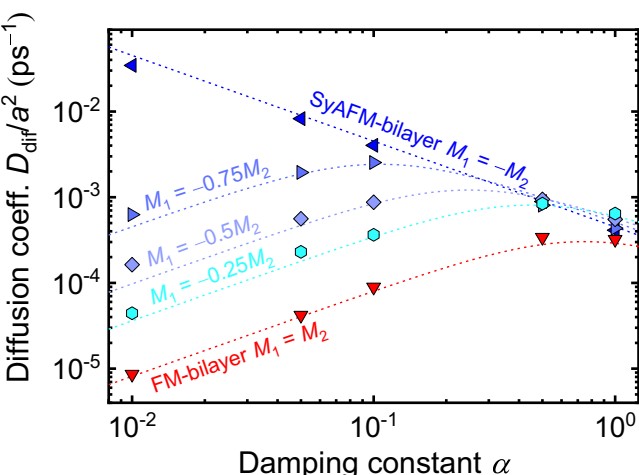

**Fig. 3 | Atomistic simulations within a model using computationally feasible small skyrmions, where *a* denotes the lattice constant[29]. a** The temperature dependence of the diffusion coefficient. The same color code as in Fig. 2a is used. **b** The Gilbert damping constant dependence of the diffusion coefficient. The atomistic simulation data is compared with the analytical description established in Eq. (1), which describes the numerical results well.

effective Thiele equation (see Supplementary Note 4),

$$D_{\text{dif}} = k_{\text{B}}T \frac{\alpha\Gamma}{\alpha^2\Gamma^2 + G^2(1 - m_{\text{Com}})^2} \quad (1)$$

where $k_{\text{B}}$, $\alpha$, $\Gamma$, and $G$ represent the Boltzmann constant, the Gilbert damping constant, the dissipative constant for the spatial derivatives of the magnetization, and the absolute value of the gyrotropic term, respectively. Equation (1) differs from the expression for the diffusion coefficient derived earlier[26] by a scaling with the effective gyrotropic force (the compensation-weighted gyrotropic term). Based on ref. 28, we expect a change in this term to heavily impact the diffusion coefficient: Increasing the compensation can suppress the microscopic thermal gyrotropic motion that arises from the finite effective topological charge. As a consequence, the SyAFM skyrmion shows a higher diffusion coefficient compared to its FM counterpart as shown in Fig. 3a. Hence, the behavior crucially depends on the interlayer exchange coupling (that causes both skyrmions to move as a single entity) and the respective saturation

magnetizations (which determine the compensation of the layer magnetizations).

The most remarkable feature is the distinctly and qualitatively different role of the Gilbert damping constant for different gyrotropic contributions. Figure 3b shows the damping dependence of the diffusion coefficient at a fixed temperature. It has been known that FM skyrmions show a counterintuitive friction dependence of the diffusive motion[26], i.e., decreasing the damping constant (friction) leads to decreasing diffusion, which is opposite to the behavior of condensed matter particles such as colloids. The fully-compensated SyAFM skyrmion, however, can retrieve the conventional behavior owing to the effective gyrotropic term being zero. Hence, if the system possesses a low damping constant, it is expected that the effective gyrotropic force significantly and qualitatively affects the diffusive motion of magnetic skyrmions via the modulation of the compensation.

On the other hand, coming back to the compensation dependence of the diffusion coefficient, the steeper slope with increasing compensation in the semi-logarithmic plot in Fig. 2b cannot be described within this simple model. Here we need to take pinning effects into account, described via an Arrhenius law, which is a critical factor for a typical sputtered magnetic film, based on the observed linear temperature behavior of the $D_{\text{dif}}$ on a semi-logarithmic scale as previously shown for FM skyrmions in the single FM layer[22–24].

Considering pinning effects, the temperature dependence of the skyrmion diffusion can be estimated as

$$D_{\text{dif}} = D_0 k_{\text{B}} T \exp(-\Delta E/k_{\text{B}}T) \quad (2)$$

with $\Delta E$ being the depinning energy of the magnetic skyrmion. If the thermal energy is much lower than this depinning energy, the skyrmion diffusion corresponds to Kramer's escape problem where the exponential factor accounts for the (low) probability that the skyrmion can overcome $\Delta E$ and escape to the next pinning site. As long as the thermal energy is much larger than the depinning energy, the free diffusive motion is recovered. From comparison with Eq. (1), it follows that $D_0 = \alpha\Gamma/[\alpha^2\Gamma^2 + G^2(1 - m_{\text{Com}})^2]$. $D_0$, as given in the Arrhenius law in Eq. (2), can also be interpreted as the attempt frequency for overcoming the energy barrier. As such, a higher $D_0$ for AFM skyrmions as compared to those in FM systems can be understood on the same grounds as the increased attempt frequency found for the superparamagnetic behavior of AFM nanoparticles[46].

Taylor expanding the Arrhenius law around the skyrmion's activation temperature $T_0$, above which the magnetic skyrmion exhibits the diffusive motion, provides

$$\ln\left[D_0 k_{\text{B}} T \exp(-\Delta E/k_{\text{B}}T)\right] \approx \ln(D_0 k_{\text{B}}T) - 2\frac{\Delta E}{k_{\text{B}}T_0} + \frac{\Delta E}{(k_{\text{B}}T_0)^2} k_{\text{B}}T \quad (3)$$

Therefore, the linear slope of $D_{\text{dif}}$ in the semi-logarithmic plot underlines that the skyrmions are still in the pinning-dominated diffusion regime. Also, importantly the slope has implications on the depinning energy for the magnetic skyrmion, which reveals that $\Delta E$ increases monotonically with the compensation ratio. We note that the averaged-depinning energy of the magnetic skyrmions is connected to the pinning of the domain wall surrounding the core domain rather than the core itself[47], that scales with the cross-sectional area of the skyrmion[48]. Thus, it is expected that the depinning energy roughly scales with the size (radius) of the skyrmion (not the area), which reads

$$\Delta E = \Delta E(R_{\text{sk.ave}}(T)) \approx \varepsilon R_{\text{sk.ave}}(T) \quad (4)$$

where $R_{\text{sk.ave}}$, and $\varepsilon$ represent the averaged radius, and the depinning energy density, respectively. Note that Eq. (4) is solely based on Fig. 2b, Fig. 4, and the previous observations[47,48]. The size dependence of the depinning energy leads to stronger pinning for bigger skyrmions

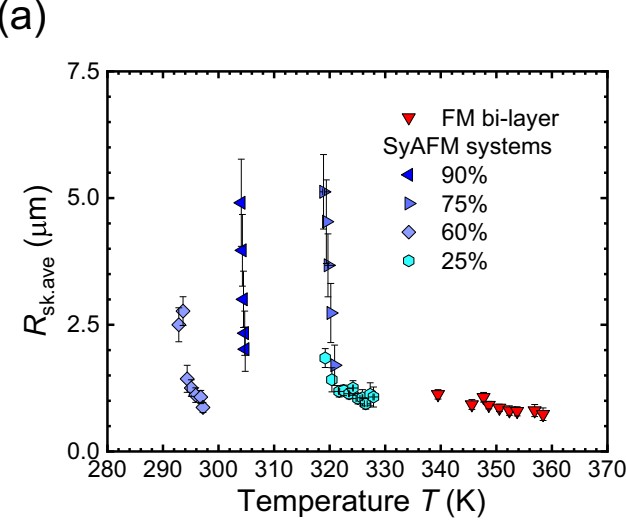

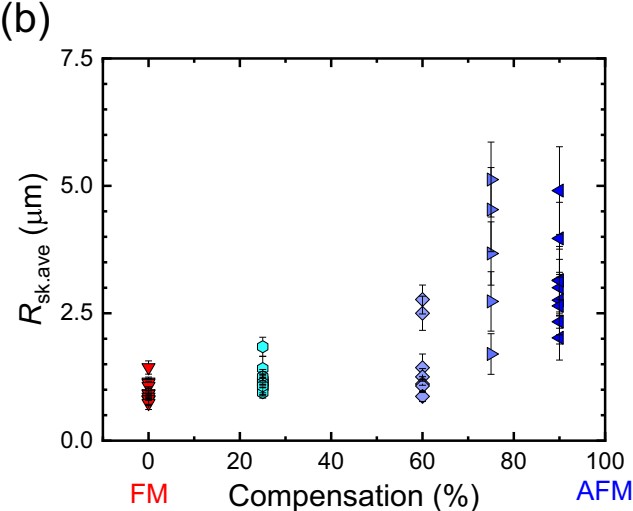

**Fig. 4 | Characterization of skyrmion size as a function of temperature. a** The temperature dependence of the average skyrmion radius is shown. The same color code as in Fig. 2a is used. **b** The magnetic compensation dependence of the average skyrmion radius, is shown for samples where diffusive motion is observed.

resulting in a lower diffusion coefficient as already observed experimentally[22]. Equation (4) demonstrates that the skyrmion size needs to be considered in order to clarify the origin of the steeper slope with varying the compensation ratio.

### Temperature dependence of the skyrmion size
Based on our theoretical analysis, we need to explore the temperature dependence of the SyAFM skyrmion size in order to investigate the systematic increase of the slope (see "Methods" and Supplementary Note 5). Figure 4a shows the temperature dependence of the skyrmion size in the temperature range where diffusive motion is observed. Intriguingly, the skyrmion size changes drastically as a function of temperature (see Supplementary Movies 2 and 3, and Supplementary Note 5). The trend is significant for the highly compensated state in line with existing theory[36], which predicts a strong temperature dependence. As shown in Fig. 4b, we find that the skyrmion size increases with the compensation, which in turn leads to an increased pinning energy based on Eq. (4). Therefore, we can conclude that the increasing slope of the diffusion coefficient is mostly attributed to the size-related pinning effect.

Whereas the effect explains well the systematic increase of the slope of the diffusion coefficient, the drastic size modification cannot

be described easily by the existing theories[36,49,50]. Assuming reasonable scaling factors of magnetic anisotropy[51,52], exchange stiffness[53,54], and interfacial DMI[52–54], these theories predict that size increases at higher temperatures, which contradicts our experimental observation so that to explain this, one needs to study in a future work the temperature dependence of the individual parameters and their impact on the skyrmion size. Also, the larger size at highly compensated states is different from the previous experimental observation in a Pt/Co/Ru/Pt/Co/Ru/Pt system[38]. One possible reason is the large difference in magnitude of the interfacial DMI. In our system, the value is too small to stabilize the skyrmion at ultra-small size, e.g., less than 10 nm, via the DMI-dominated mechanism[55] (-0.01–0.06 mJ/m² is found from Brillouin light scattering; see Supplementary Note 6) owing to the amorphous nature of the system or possibly the opposite sign of the DMI between top and bottom FMs. Another possible reason is the interlayer stray field effect which has often been neglected in the theory. Indeed, the recent experimental observation indicates that the interlayer dipole interaction could be a key enabler for stabilizing the topological spin structures[56] as Fig. 4b implies. Note that the topology of the skyrmions is fixed to be 1 due to the antiferromagnetic coupling domain walls forming flux closure[57], which is corroborated by the current-induced coherent and isotropic collective motion of domains[14] (see "Methods", Supplementary Movies 4 and 5). Nevertheless, the chirality hinges on the compensation ratio owing to competing energies for the interfacial DMI, the interlayer exchange coupling, and the interlayer stray field (see Supplementary Note 7). In our stacks, the dipole interactions play a key role in the stabilization of skyrmions whereas their influence on the diffusion is negligible as shown in previous work[22].

### Size dependence of the diffusion coefficient
Finally, we disentangle the size-related pinning influence to identify the pure influence of the effective gyrotropic force on the diffusion. This is realized in Fig. 5a by plotting the size dependence of the diffusion coefficient on a semi-logarithmic scale at a specific fixed $T$, incorporating reference data[22] as well. The reference was previously obtained in a Ta/CoFeB/MgO system for FM skyrmions for which their diffusive motion data is available around 300 K. On the basis of Eq. (3) with $\Delta E$ as in Eq. (4), the slope and the intercept in Fig. 5a can be related to the depinning energy density and the intrinsic diffusion coefficient, respectively. The depinning energy density also includes the non-homogeneous contribution of the energy landscape stemming from the spatial variation of all relevant magnetic parameters such as the magnetic anisotropy.

Figure 5b indicates that the compensation dependence of $D_0$ normalized to that of FM skyrmions (the intercept divided by $T$ to account for the temperature difference), represents the intrinsic diffusion coefficient reflecting the effect of the microscopic gyrotropic force on the thermal dynamics of the skyrmion, where $D_0^{FM}$ denotes the averaged $D_0$ for FM skyrmions in Fig. 5a. We clearly observe an increase in the diffusion coefficients with increasing the compensation and an approximately 10–30 times larger diffusion coefficient for the 90% compensated SyAFM skyrmions compared to that of the FM skyrmions. To verify the qualitative consistency, we calculate the compensation ratio dependence of $D_0$ based on Eq. (1) which is shown by dashed lines in Fig. 5b. We note that skyrmion radius $R_{sk.ave}$ and domain wall width $\delta$ would affect the intrinsic diffusion coefficient via the dissipative term $\Gamma$ that is proportional to $R_{sk.ave}/\delta$[11]. Hence, we could access only the averaged $D_0$ experimentally for varying $R_{sk.ave}$ given that $\Gamma$ is put to be constant to obtain the intrinsic $D_0$. Thus, various $R_{sk.ave}$ and $\delta$ are accounted for in the calculation, which are experimentally obtained for each stack (see "Methods" for more details). We find that our calculation reproduces the monotonic increasing of experimental compensation dependence, where we emphasize that only changing $\Gamma$ with

assuming zero compensation cannot describe the 10-30 times larger diffusion coefficients. It is worth noting that, at the same compensation ratio, smaller skyrmions exhibit a larger $D_0$ value due to the dissipative term $\Gamma$. This observation indicates that the enthalpy-entropy compensation, in which larger skyrmions are predicted to show a greater $D_0$ as a result of larger depinning energy[58], cannot provide a sufficient explanation for our experimental results. Therefore, the enhanced diffusion coefficient has to be primarily ascribed to the suppression of the effective gyrotropic force in the highly compensated SyAFM system.

## Discussion

We have realized skyrmion diffusion in low pinning synthetic antiferromagnets, which in contrast to amorphous ferrimagnets or crystalline antiferromagnets with strong pinning allow us to experimentally probe predictions of the drastic impact of the microscopic gyrotropic force, arising from the topology, on the diffusion.

Previously it has been found to be very challenging to observe the stochastic dynamics of topological spin textures in antiferromagnetic systems, and thus the application for antiferromagnetic skyrmions has been limited to deterministic devices. However, antiferromagnets have been predicted to be intrinsically preferable for probabilistic spintronic devices including Brownian computing rather than for conventional deterministic devices, which require reproducible dynamics owing to enhanced stochasticity and thus rapid diffusion[28,36,46]. By designing multilayer systems of amorphous CoFeB with low pinning, we have successfully demonstrated thermally-activated antiferromagnetically coupled skyrmion diffusion with a more than 10 times larger diffusion coefficient compared to the conventional ferromagnetic counterparts. Using our developed analytical formula, we can qualitatively describe to result from the suppression of the effective gyrotropic force. Whereas we have intentionally used relatively large skyrmions, which allows for the observation via a MOKE microscope, the enhancement of the diffusion coefficient will be more significant for DMI-stabilized smaller skyrmions, owing to a smaller dissipative term $\alpha\Gamma$. Furthermore, the DMI-stabilized AFM skyrmions would maintain sufficient thermal stability even at single-digit size scales[55], which suggests the potential for a broader working temperature range in highly compensated states contrary to dipole-stabilized large skyrmions. Consequently, they offer promising implications for device scalability.

In summary, our findings that provide crucial insights into the thermally-activated dynamics of the topological objects in antiferromagnet systems would enable scalable effective unconventional computing using antiferromagnetic systems for which a probabilistic operation based on fast diffusion is key.

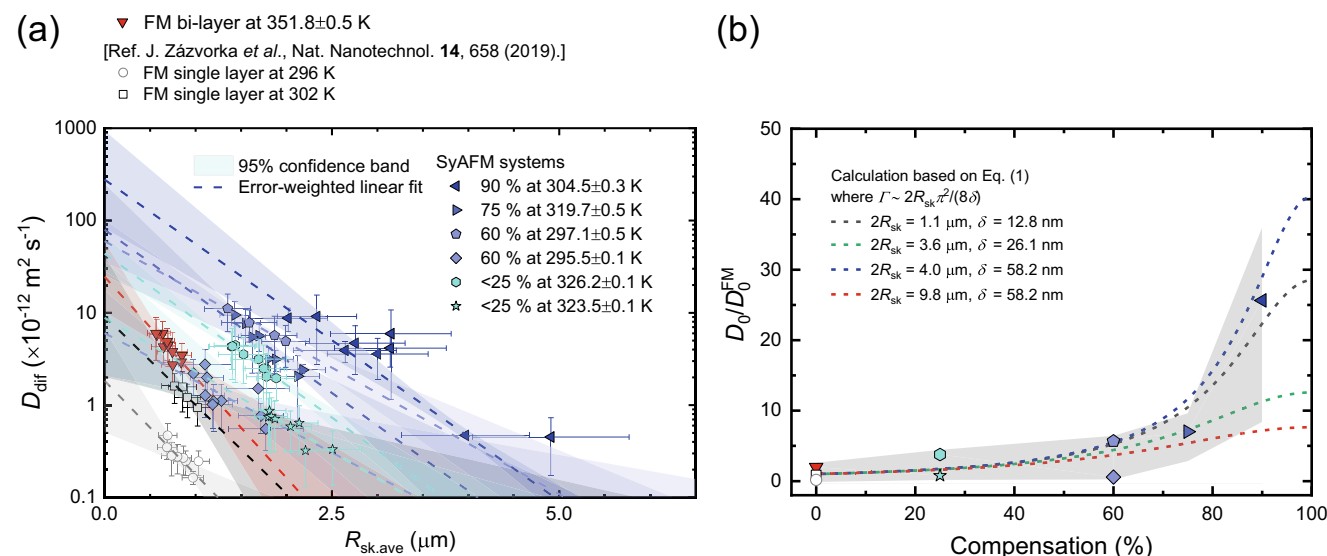

**Fig. 5 | The impact of the effective gyrotropic force on the diffusion coefficient and depinning energy of the systems. a** The averaged-size dependence of the diffusion coefficient at a fixed temperature. The color code used is consistent with Fig. 2a. The skyrmion size was controlled by an out-of-plane magnetic field. The dashed line and shaded area represent the error-weighted linear fit, and the 95% confidence band calculated from the fit, respectively. **b** The dependence of the $D_0/D_0^{FM}$ ratio on the compensation, where $D_0^{FM}$ denotes the averaged $D_0$ for FM skyrmions in Fig. 5a. The value of $D_0$ is obtained by dividing the intercept by $T$ to account for the temperature difference. The same symbols and color coding as in Fig. 5a are employed. The shaded area corresponds to the error bar, calculated using the standard deviation. The broken line displays the theoretical calculation based on Eq. (1), using the experimentally obtained parameters for the SyAFM system with 90%, 60%, and 0% (FM bi-layer) compensation, where $\Gamma \sim 2R_{sk.ave}\pi^2(8\delta)^{-1}$ based on ref. 11.

## Table 1 | Summary of FM thickness for each compensation ratio

| | $t_{CFB1}$ (nm) | $t_{FCB1}$ (nm) | $t_{CFB2}$ (nm) | $t_{FCB2}$ (nm) |
|---|---|---|---|---|
| FM bi-layer (covered by Ru, 1.60 nm Ir) | 0.50 | 0.35 | 0.50 | 0.35 |
| $m_{Com} = 25\%$ (covered by HfO$_x$, 1.20 nm Ir) | 0.45 | 0.40 | 0.45 | 0.40 |
| $m_{Com} = 60\%$ (covered by Ru, 1.20 nm Ir) | 0.45 | 0.40 | 0.45 | 0.40 |
| $m_{Com} = 75\%$ (covered by Ru, 1.20 nm Ir) | 0.43 | 0.48 | 0.43 | 0.48 |
| $m_{Com} = 90\%$ (covered by Ru, 1.20 nm Ir) | 0.43 | 0.50 | 0.43 | 0.45 |

**Table 2 | Summary of magnetic parameters used for the calculation in Fig. 5b**

| | Thickness (nm) | $M_S$ (T) | $\mu_0 H_K^{eff}$ (mT) | $A_S$ (pJ/m) | $\delta$ (nm) | $\alpha$ |
|---|---|---|---|---|---|---|
| FM bi-layer | 1.70 | 1.0 | 147 | 12.1 | 12.8 | 0.022 |
| $m_{Com}$ = 60% | 1.70 | 0.4 | 44 | 5.6 | 26.1 | 0.022 |
| $m_{Com}$ = 90% | 1.82 | 0.1 | 45 | 6.2 | 58.2 | 0.022 |

## Methods

### Film preparation

The samples were prepared using a Singulus Rotaris sputtering machine with a base pressure better than $3 \times 10^{-8}$ mbar. Employing the Singulus Rotaris deposition system, we can tune the thickness of the layers with a high accuracy (reproducibility better than 0.01 nm). Table 1 summarizes the FM thickness (extracted from the deposition rate calibrated from X-ray reflectivity measurements) used for each compensation ratio. We utilized the different composition of CoFeB, with Co-rich CoFeB ($Co_{0.6}Fe_{0.2}B_{0.2}$) and Fe-rich CoFeB ($Co_{0.2}Fe_{0.6}B_{0.2}$) playing distinct roles in the system. Whereas Co-rich CoFeB acts as a source of the interfacial magnetic anisotropy and DMI, Fe-rich CoFeB is deposited to render the systems close to amorphous to reduce pinning and evoke diffusive motion. In our SyAFM system, the material layers deposited on top of the FMs are different for the bottom and top FMs to break the spatial inversion symmetry (Ir, or Ru), causing a difference in the magnetic dead layer[59], which provides a not fully-compensated state under the condition $t_{CFB1} + t_{FCB1} = t_{CFB2} + t_{FCB2}$. Due to Ir being a heavy metal, the bottom FMs typically have lower magnetic moment for which we easily obtain the preferable condition that is $M_2 > M_1$, since the Kerr imaging is a surface sensitive technique. This is supported by the fact that even when using the completely same thickness for the bottom and top FMs, we obtained a different compensation by using different capping layers (see 25% and 60%). For the 25% compensation SyAFM the top FM is covered by $HfO_x$, exhibiting a lower compensation compared to the SyAFM covered by Ru owing to a thinner dead layer for the top FM. Note that even though the magnetic proximity effect[60,61] could also cause such deviation of magnetic moments, the relationship becomes reversed ($M_1 > M_2$) as the bottom seed Pt layer[62] has larger proximity-induced moments when the effect is significant. This is not the case though for our system.

### Magnetic parameters

The saturation magnetization $M_S$ and compensation ratio of magnetic moments are determined from the $m$-$H$ curve obtained from a superconducting quantum interference device and a vibrating sample magnetometer. The effective magnetic anisotropy field, the exchange stiffness $A_S$, and the magnitude of DMI are determined from the spin-wave dispersion relation obtained using the Brillouin light scattering method[63], where we also determined the damping constant $\alpha$ to be 0.022 assuming a gyromagnetic ratio of 28 GHz $T^{-1}$, which is consistent with previous work[59,64]. For the Gilbert damping constant, the line width of a spin-wave mode close to the ferromagnetic resonance is measured as a function of the magnetic field to be able to separate it from the inhomogeneous broadening. Following an analytical approximation formula of the dissipative tensor[11], we roughly estimated the dissipative term $\alpha\Gamma$ as well as the gyrotropic term $G$ for the SyAFM system with $m_{Com}$ = 90, 60, and 0% (FM bi-layer system), reproducing consistent results based on Eq. (1) in Fig. 5b. For the calculation, we assume domain wall width $\delta$ to be constant whereas $R_{sk.ave}$ changes at the fixed $T$ as skyrmion size was controlled by changing magnetic fields. Table 2 summarizes the parameters used for the calculation in Fig. 5b, in which $\delta$ is also calculated using those parameters[65]. The interlayer exchange coupling $J_{int}$ is estimated by using the relation[66], $J_{int} = -M_S H_S t_T$, where $H_S$ and $t_T$ are the saturation

field and the total ferromagnetic thickness, respectively. For 90% compensation, $J_{int}$ is determined to be ~0.14 mJ/m² (the used parameters are $M_S$ = 1 T, $\mu_0 H_S$ = 100 mT, and $t_T$ = 1.82 nm).

### Experimental setup

A commercial Evico GmbH MOKE microscope combined with three light-emitting diodes (LEDs) providing white light with a continuous wavelength spectrum was used in a polar configuration for detecting the magnetic domain state. The differential image between uniform and non-uniform states was recorded to obtain better contrast. The frame rate used for recording movie was 16 frames per second (fps) corresponding to a time resolution of 62.5 ms. To be able to apply both out-of-plane and in-plane magnetic fields simultaneously for controlling skyrmion density[67], the electromagnetic coil was custom made at Johannes Gutenberg-Universität Mainz. This coil was calibrated by a magnetometer with a Hall probe. For the Peltier module, it was confirmed that the temperature ranging from 290 to 360 K was stable during the measurement. The temperature was monitored by a Pt100 resistive temperature sensor which was placed on top of the Peltier element next to the sample.

### Skyrmion tracking and size evaluation

The Python module TrackPy was used to track skyrmions[68,69]. For full statistics, multiple magnetic skyrmion movies were recorded for approximately 2 to 3 min, of which 2000 frames were used for tracking. Note that since we use a continuous film, magnetic skyrmions move in and out of the view; also the trackable time for the diffusive motion is limited. The positions of skyrmions were detected using appropriate conditions[70,71], and then the linking function was employed to obtain the skyrmion trajectories during diffusive motion as can be seen in Supplementary Movie 1. The functionality of the used algorithm was confirmed manually using a few skyrmions, by checking the success of the tracking. In order to focus on extracting the intrinsic diffusion coefficient, individual skyrmions were measured without any confinement which leads to the anomalous diffusive motion[72,73]. The error bar of the diffusion coefficient was calculated as the standard deviation (s.d.) of the MSDs. For the size evaluation, the radius of gyration of the Gaussian-like profile was employed to take into account the deformation of skyrmion over time[36] (see Supplementary Note 5). The error bar was also calculated as the s.d. of the skyrmion sizes. Note that the topology dependence of Brownian gyromotion as demonstrated in ref. 24 was not observed in this work due to limited statistics.

### Current-induced domain motion

The continuous magnetic film (SyAFM system with 90% compensation) was first cleaved into small pieces with a width of 3 mm. After that, a contact pad consisting of Cr(5)/Au(30) was constructed through the liftoff process to obtain uniform current distribution in the film. Any baking/annealing processes were avoided to preserve the properties of the film. The resistance was 82 Ω and 8 V with a 10 ms pulse width and a pulse repetition rate of 1 s, resulted in a current density of ~3.2 × 10⁹ A/m², which was applied in Supplementary Movies 4 and 5. Positive and negative pulses (regarding the bottom side being the ground) were used for Supplementary Movies 2 and 3, respectively. To avoid mixing diffusive motion during drift motion, we investigated the collective domain motion to check whether the chirality is fixed throughout the film. As can be seen in Supplementary Movies 4 and 5 (note that the magnification is different from the other movies), it is unambiguously shown that the domain is collectively displaced along the current flow direction (opposed to electron flow direction), indicating that the stable left-handed Néel-type or Bloch-Néel hybrid-type domain walls in the film were moved by a SOT generated from the Pt heavy metal layer with the positive spin Hall angle of Pt[74]. The obtained velocity is 3.1 ± 0.3 mm s⁻¹. We also find that the domain wall chirality depends on the compensation ratios using the experimentally

obtained parameters from left-handed Néel-type to right-handed Néel-type, as well as the hybrid Néel for FM bi-layer systems (see Supplementary Note 7).

## Atomistic simulations of thermal skyrmion motion

We perform atomistic spin simulations to compare our results for the Brownian motion of skyrmions with the analytical theory we establish in Supplementary Note 4. To be able to realize these simulations with reasonable computational effort, we use toy model parameters instead of the magnetic parameters for the experimentally investigated SyAFM and FM bi-layer stacks. In such a toy model the skyrmions are of the order of a few nanometers and their thermal motion can be observed using finite temperature spin model simulations, while this is challenging for the large skyrmions in the experimentally used stacks. Besides, the experimental timescales of seconds can be computed neither using atomistic spin models nor micromagnetics.

For the simulations, we use an atomistic spin Hamiltonian of the form

$$H = -\frac{1}{2}\sum_{i,j} J_{ij}\mathbf{S}_i \cdot \mathbf{S}_j - \frac{1}{2}\sum_{i,j} \mathbf{D}_{ij} \cdot \left(\mathbf{S}_i \times \mathbf{S}_j\right) - \kappa \sum_i S_{i,z}^2 \quad (5)$$

where the first term is the Heisenberg exchange, the second term is the Dzyaloshinskii–Moriya interaction (DMI) and the third is the uniaxial anisotropy. Note that it is assumed that the z-axis corresponds to the out-of-plane direction. The $\mathbf{S}_i$ denote the normalized magnetic moments which are located at each lattice site. The lattice structure is assumed to be a simple cubic system with dimensions $2 \times N \times N$ in order to describe a bi-layer structure. The isotropic Heisenberg coupling within each monolayer has a value of $J_{intra} = 10$ meV and the Heisenberg coupling between the layers is $J_{inter} = \pm J_{intra}$, depending on whether a FM bi-layer or a SyAFM is simulated. Note that for the sign convention used in Eq. (5) a positive or negative Heisenberg coupling corresponds to FM or AFM coupling, respectively. The DMI within each layer is assumed to be of an interfacial type and hence the DMI vectors $\mathbf{D}_{ij}$ are within the x−y plane. They are orthogonal to the vector connecting the lattice site $i$ and $j$ and their absolute value is $D = 3$ meV. The DMI between the two monolayers is assumed to be zero. The uniaxial anisotropy constant is set to $\kappa = 1.5$ meV.

To investigate the dynamics, we solve the stochastic Landau-Lifshitz-Gilbert equation of motion[45]

$$\frac{\partial \mathbf{S}_i}{\partial t} = \frac{-\gamma}{(1+\alpha^2)m_i} \mathbf{S}_i \times \left(\mathbf{H}_i + \alpha \mathbf{S}_i \times \mathbf{H}_i\right) \quad (6)$$

with the Gilbert damping constant $\alpha$, the gyromagnetic ratio $\gamma = 1.76 \times 10^{11}$ rad s$^{-1}$ T$^{-1}$ and the effective field $\mathbf{H}_i = -\partial H/\partial \mathbf{S}_i + \zeta_i$, where $\zeta_i$ are thermal fluctuations in the form of Gaussian white noise. The saturation magnetic moments in the two monolayers are given by $m_1 = M_1 a^2$ and $m_2 = M_2 a^2$, where $a$ is the lattice constant.

The simulations are performed via a GPU-accelerated implementation of Heun's method[45] with a fixed time step of 0.1 fs and a system consisting of $2 \times 64 \times 64$ spins and periodic boundary conditions are assumed. Initially, a SyAFM or FM skyrmion is placed in the center of the system and thermalized at finite temperature. Subsequently, its position is tracked every 5 ps over 500 ps by using an adaption of the algorithm in Appendix A of ref. 26. For each set of parameters (temperature, damping and compensation) 100 simulations are performed, in order to provide sufficient statistics. An example of a simulation for the SyAFM skyrmion diffusion is available in Supplementary Movie 6.

## Data availability

The data supporting the findings of this work are available from the corresponding authors upon reasonable request.

## Code availability

The computer codes used for data analysis are available upon reasonable request from the corresponding author.

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

## Acknowledgements

This project has received funding from the Deutsche For- schungsgemeinschaft (DFG, German Research Foundation) Project No. 403502522 (SPP 2137 Skyrmionics). The group in Mainz acknowledge funding from the DFG through SFB TRR 146 as well as financial support from the Horizon 2020 Framework Programme of the European Com- mission under FET-Open Grant No. 863155 (s-Nebula) and Grant No. 856538 (ERC-SyG 3D MAGiC). M.W. and U.N. in Konstanz acknowledge funding from the DFG through the SFB 1432. This work was partly sup- ported by JSPS Kakenhi (No. 23K13655). The groups in Mainz and Kai- serslautern acknowledge funding from TopDyn and SFB TRR 173 Spin+X (project A01 and project B01 #268565370). G.J. and M.A.S. acknowl- edge funding from the European Union's Horizon 2020 research and innovation program under the Marie Skłodowska-Curie Grant Agree- ment No. 860060 "Magnetism and the effect of Electric Field" (Mag- nEFi). This work was supported by the Max Planck Graduate Center with the Johannes Gutenberg-Universität Mainz (MPGC). T.D. gratefully acknowledges the help and advice of many members including the technicians of the Kläui group, and financial support by the Canon Foundation in Europe. N.K. and M.K. gratefully acknowledge financial support by the Graduate School of Excellence Materials Science in Mainz (Mainz, GSC266). J.Z. acknowledges support from Charles Uni- versity (Grant No. PRIMUS/20/SCI/018).

## Author contributions

T.D., M.W., U.N., and M.K. initiated, designed, and supervised the project. T.D., F.K., and M.A.S. designed and prepared the stack structure. T.D. prepared the experimental setup together with K.R., F.K., and N.K.; T.D. and M.A.S. conducted magnetization measurements with technical support from G.J.; T.D. measured the diffusive and current-induced dynamics of magnetic skyrmions. T.D. and Y.G. analyzed the experi- mental data for the diffusive dynamics. A.S. conducted the Hall mea- surement. T.B., M.R., and P.P. measured and evaluated the Brillouin light scattering spectra. M.W. derived the analytical formula and performed the atomistic simulation. T.D., N.K., and M.W. drafted the manuscript with help of U.N. and M.K. All authors discussed the results and com- mented on the manuscript.

## Funding

## Competing interests

The authors declare no competing interests.
