## [Peer Review File · Nature Communications]

Reviewers' Comments:

Reviewer #1:

Remarks to the Author:

Report of the manuscript by T. Dohi et al.

After reading this revised manuscript, I can see a substantial improvement as compared with the early version that was submitted somewhere else. I appreciate authors take these great efforts. Below are my specific comments that authors should take into consideration before I can recommend its publication.

1: While I can see improvement of the displayed items, I am afraid that they are not adequately yet, especially for Figure 2. Figure 5a is also a mess.

2: Regarding to the temperature dependent diffusion coefficient, I have a concern with respect to the anisotropy constant. This parameter determines the energy barrier and hence could contribute to the thermally activate diffusion. Can authors comment on this issue?

3: Since the effective topological charge is not fully compensated, I am curious to see if there exists a topology dependent brownian motion? I understood this could be experimentally challenging. Can authors at least comment on this aspect based on the micromagnetic simulation, or atomistic simulation?

4: I am also curious about the drastic change of the skyrmion size as a function of temperature? What are the controlling factors that determine the size of nearly compensated skyrmions in the current material system? How they change as a function of temperature? This needs a careful examination.

5: English needs some attention.

After addressing these issues, I can suggest its publication in Nature Comm.

Reviewer #2:

Remarks to the Author:

In this work, the authors experimentally and theoretically study the thermally-activated diffusive motion of skyrmions in a synthetic antiferromagnetic system with low pinning. By tuning the compensation ratio of magnetic moments in the magnetic layers, they observe a more than 10 times larger diffusion coefficient for highly compensated antiferromagnetically-coupled skyrmions compared to the conventional ferromagnetic counterparts. The diffusive motion of skyrmions is a current hot topic in spintronics, as well as the results of this work may be potentially useful since future technological applications based on magnetic skyrmions require a full understanding of the skyrmion dynamics in the presence of thermal fluctuations. In general, this manuscript can be accepted after the following issues are clarified and corrected.

Main issues:

1) In Figure 2(d), one can see that the temperature dependence of the diffusion coefficient for the stacks lies in different temperature ranges. Is it possible to show the $D_{\text{dif}}-T$ relationship for the stacks with different compensation ratio in the same temperature range?

The authors claim that the measurable temperature range is limited due to constraints regarding the skyrmion nucleation, stability and the time resolution of their imaging setup. This may be right to a certain extent. However, I still would like the authors to argue more about this point or propose some potential methods to solve this problem, since the working window (i.e., the measurable temperature range) for each stack is very narrow, which is not beneficial for future applications.

In addition, I wonder to know why the measurable temperature range of the system with a compensation ratio of 60% is not between that of the systems with the compensation ratio of 75% and 25%, but is independently located on the far left.

2) Regarding the compensation dependence of the diffusion coefficient, the authors conduct a theoretical analysis based on Eqs. (2-4). To understand the steeper slop with increasing compensation ratio, in my opinion, finding the relationship between D_{dif}/T [or $\ln(D_{\text{dif}}/T)$] and m_{com} (or the mentioned ΔE) is the most intuitive and persuasive way. It is not

straightforward to draw the conclusion that " ΔE increases monotonically with the compensation ratio" shown on page 10, line 207, from the current Eqs. 2 and 3. Therefore, can the authors provide an equation with the form of " $D_{\text{dif}}/T = \dots \Delta E$ "?

3) I understand that the authors using the collective domain motion to check the chirality of the spin texture in this system. I would like to know how to distinguish the skyrmion, skyrmion bubble and magnetic bubble in experiments.

4) In the experimental section, the authors claim that the DMI is very weak in their system and the dipolar coupling effects play a key role in the stabilization of skyrmions. However, in the simulation, the dipole-dipole interaction is lacking in the spin Hamiltonian. The authors should clarify this point.

And also, can the authors provide one Figure or one Video to show the diffusive motion of SyAFM skyrmions in the simulation?

Other issues:

1) On page 5, lines 102-104, there are many variables denoted by 'm' or 'M'. Please give the definition for each. Especially, do 'm_{CFB}' and 'm_{FCB}' indicate the saturation or unit magnetization of magnetic layers?

2) Please double check the main text and the supplementary information to eliminate typos and language deficiencies. For example, the last column of Table 2 should be the damping constant, denoted by α instead of A, and there is a wrong citation of Figures in Supplementary Note 3 [see "Based on the peak indicating the interlayer exchange coupling field in Fig. 2(b)"].

3) To improve the visibility of Figures and avoid any potential confusion, I suggest that the authors can make some modifications to the Figures that include too many things, such as Figure 2 and Figure 5.

4) Please make sure the labelling of variables is consistent. For instance, in the experimental section, the diffusion coefficient is denoted by ' D_{dif} ', while in the theoretical part it is denoted by D/a^2 (see Figs. 2 and 3). The units of temperature used in experiment and theory are also different.

Reviewer #3:

Remarks to the Author:

Referee report on the manuscript entitled "Enhanced thermally-activated skyrmion diffusion with tunable effective gyrotropic force" by Takaaki Dohi et al.

The topic of the manuscript is timely and of interest. Recently, it is clear that the conventional application of magnetic skyrmions such as memory are not competitive and then there is a great effort in looking for unconventional application where a skyrmion based technology can perform better. The work presented here is in context and it is a complete description of application of magnetic skyrmions for stochastic computing. Theory and experimental data well support the conclusions of the work.

The idea of stochastic computing with skyrmions is not new, some of the authors were pioneering on this. Indeed, the stabilization of SyAF skyrmions is by itself an hot topic.

I have enjoyed the reading and I'm positive with the publication. However, the following points should be addressed before.

1. The discussion "the SyAFM skyrmion shows a higher diffusion coefficient compared to its FM counterpart as shown in Fig. 3a. Hence, the behavior crucially depends on the magnetic parameters" is too generic. Here the key ingredients are the interlayer exchange coupling and the effective magnetizations.

2. I suggest to add data on skyrmion motion driven by the current and add a comparison in the

text. This can be used to identify the pinning-dominated diffusion regime.

3. About the skyrmion size dependence on temperature the authors can refer to the papers: Rózsa, et al Phys. Rev. B 96, 094436, 2017 and Tomasello et al Phys. Rev. B 97, 060402(R) 2018 to verify the scaling of the DMI coefficient to explain the size effect. Considering that D is small, it is expected that also the scaling of other parameters can be important for the size calculation.

4. The data on BLS for evaluating the DMI should be included. Please include also the parameters used for the estimation.

5. The discussion about the chirality should be improved. It is not completely clear to me. I suggest adding a 2d phase diagram of the chirality considering interlayer exchange and composition.

6. It is not clear which is the peak the authors are referring in Supplementary Fig. 1 (b).

Response to Reviewer #1

We are grateful to the reviewer for the conscientious reading of our manuscript and the helpful suggestions. We are glad to hear the positive appreciation of our work and we have revised the manuscript according to the comments and incorporated the suggestions. Our responses are listed below.

Reviewer's comment

Report of the manuscript by T. Dohi et al. After reading this revised manuscript, I can see a substantial improvement as compared with the early version that was submitted somewhere else. I appreciate authors take these great efforts. Below are my specific comments that authors should take into consideration before I can recommend its publication.

Our reply

We are pleased to learn that the reviewer has recognized the significantly improved quality of our manuscript and is inclined to support its publication in Nature Communications. We also appreciate the reviewer's valuable suggestion for further enhancing the quality of our work.

Reviewer's comment

1. 1: While I can see improvement of the displayed items, I am afraid that they are not adequately yet, especially for Figure 2. Figure 5a is also a mess.

Our reply

In order to improve the visibility of Fig. 2 and Fig. 5(a), we have made revisions to these figures. Specifically, with respect to Fig. 2, we have removed Fig. 2(a), (b), and (c), which were not necessary to focus on the findings of this work, as the same behavior has already been shown in previous works [J. Zázvorka *et al.*, Nat. Nanotechnol. **14**, 658 (2019); T. Nozaki *et al.*, Appl. Phys. Lett. **114**, 012402 (2019); L. Zhao *et al.*, Phys. Rev. Lett. **125**, 027206 (2020)]. Instead, we have integrated Fig. 2(a), which shows the skyrmions observed in our systems, into Fig. 1(a) to demonstrate that we have clearly observed the skyrmions. As for Fig. 5(a), we have adjusted the colors by lightening the error region and strengthening the color of the fitted line to make it more visible. Regarding Figure 5(b), we removed two solid lines from the theoretical calculation to make it easier to see, while still displaying the trend of the intrinsic diffusion coefficient, which gradually changes with the dissipative constant Γ . We believe that these revisions have significantly improved the visibility of the figures compared

to the previous versions. Please refer to the revised figures and the revised sentences corresponding to them below and if the reviewer has additional concrete suggestions to improve the accessibility of our work, we are happy to hear them and revise the figures further:

<From>

<To>

[page 29, Figure 1 (a)]

<From>

<To>

<From>

<To>

<The summary of minor changes for the corresponding sentences in the main text>

[page 29, The caption of Fig. 1]

[page 30, The caption of Fig. 2]

[page 31, The caption of Fig. 5]

[page 6, lines 116-117 in the main text]

[page 6, line 126 in the main text]

[page 6, line 132 in the main text]

[page 6, line 135 in the main text]

Reviewer's comment

2. *2: Regarding to the temperature dependent diffusion coefficient, I have a concern with respect to the anisotropy constant. This parameter determines the energy barrier and hence could contribute to the thermally activate diffusion. Can authors comment on this issue?*

Our reply

We agree with this comment. The spatial variation of all magnetic parameters (exchange, anisotropy, DMI, magnetization, etc...) contributes to the inhomogeneity of the energy landscape. As such, the anisotropy constant certainly plays a key role. However, what is relevant here is not individual contributions, but rather the full energy landscape. Individually extracting such spatially varying magnetic properties including the anisotropy constant is experimentally challenging. Nonetheless, the most important we would like to highlight here is that all the contributions relevant to the energy barrier are automatically taken into account (for equations 3 and 4) by estimating the effective pinning energy density for different material stacks. In order to emphasize this point we add the sentence as follows:

[page 12, lines 265-267 in the main text]

<From>

Fig. 5a can be related to the depinning energy density and the intrinsic diffusion coefficient, respectively.

<To>

Fig. 5a can be related to the depinning energy density and the intrinsic diffusion coefficient, respectively. **The depinning energy density also includes the non-homogeneous contribution of the energy landscape stemming from the spatial variation of all relevant magnetic parameters such as the magnetic anisotropy.**

Reviewer's comment

3. 3: *Since the effective topological charge is not fully compensated, I am curious to see if there exists a topology dependent brownian motion? I understood this could be experimentally challenging. Can authors at least comment on this aspect based on the micromagnetic simulation, or atomistic simulation?*

Our reply

The *topology dependence of Brownian motion* of skyrmions is a result of the gyrocoupling and leads to two effects present on two distinct timescales. For long times, it leads to an overall decrease of the diffusive motion (i.e. their diffusion coefficient). This is what we have investigated in our work by demonstrating the dependence of skyrmion Brownian motion on the effective gyrocoupling, and with that on the topological charge.

The second impact is visible in the short-term dynamics: the ballistic motion of skyrmions is realized on circular trajectories. This is what has been demonstrated in [L. Zhao *et al.*, Phys. Rev. Lett. **125**, 027206 (2020)]. In order to reveal this effect, one has to average over a large set of data, which would require additional simulations/experiments. It can be understood, however, from analytical calculations based on a massive Thiele equation, which is obtained by supplementing the Thiele equation with a mass term M_0V .

The topology-dependent Brownian gyromotion can be quantified using the velocity autocorrelation function, which can be calculated from the massive Thiele eq. as [M. Weißenhofer, Doctoral dissertation, Konstanz, Univ. (2022).]

$$\langle V(t) \cdot V(0) \rangle = 2 \frac{k_B T}{M_0} e^{-\frac{\alpha D}{M_0} t} \cos\left(\frac{G}{M_0} t\right).$$

The gyromotion is represented by the G -dependent cosine function. By replacing $G \rightarrow G(1 - m_{\text{Com}})$ and $M_0 \rightarrow Mm_{\text{Com}} + M_0$ – as in our manuscript – we obtain

$$\langle V(t) \cdot V(0) \rangle = 2 \frac{k_B T}{Mm_{\text{Com}} + M_0} e^{-\frac{\alpha D}{Mm_{\text{Com}} + M_0} t} \cos\left(\frac{G(1 - m_{\text{Com}})}{Mm_{\text{Com}} + M_0} t\right).$$

Using this expression, we predict a vanishing gyromotion for $m_{\text{Com}} = 1$, indicating that the ballistic motion transforms from circular to linear. The vanishing of the gyromotion is also responsible for the drastic increase in the long term – i.e. the diffusive – dynamics (see above).

Note that we assume a small, but finite contribution to the mass, denoted by M_0 , for the ferromagnetic skyrmion. This mass is absent in the calculations in the manuscript since it does not impact the long-term (diffusive) dynamics that is studied here. However, it is known from experiments [L. Zhao *et al.*, Phys. Rev. Lett. **125**, 027206 (2020).] that ferromagnetic skyrmion does, in fact, have finite mass. This is supported by theory [R. E. Troncoso and Á. S. Núñez, Annals of Physics **351**, 850

(2014).], that suggests that this mass originates from deformations (see also our earlier work in Nat. Phys. **11**, 225 (2015).), which is absent within Thiele's rigid body approach. Note that the observation of this effect is experimentally challenging. Therefore, we have added one sentence regarding that as follows.

[page 23, lines 512-513 in Method]

Note that the topology dependence of Brownian gyromotion as demonstrated in Ref. 24 was not observed in this work due to limited statistics.

Reviewer's comment

4. 4: *I am also curious about the drastic change of the skyrmion size as a function of temperature? What are the controlling factors that determine the size of nearly compensated skyrmions in the current material system? How they change as a function of temperature? This needs a careful examination.*

Our reply

Indeed, revealing the critical factors determining the size of nearly compensated skyrmions is the important issues of the field of skyrmionics. For a large, bubble-like skyrmion the radius follows this formula [S. Rohart and A. Thiaville, Phys. Rev. B **88**, 184422 (2013).]

$$R = \frac{\sqrt{\frac{A}{K}}}{\sqrt{2 \left(1 - \frac{D\pi}{4\sqrt{AK}} \right)}}$$

When we use the typical scaling behavior of the parameters [Rózsa, *et al.*, Phys. Rev. B **96** 094436, (2017).] $A = A_0 m^{1.54}$, $D = D_0 m^{1.54}$, and $K = K_0 m^3$ we get

$$R = \frac{\sqrt{\frac{A_0 m^{1.54}}{K_0 m^3}}}{\sqrt{2 \left(1 - \frac{D_0 m^{1.54} \pi}{4\sqrt{A_0 m^{1.54} K_0 m^3}} \right)}} = \frac{\sqrt{\frac{A_0}{K_0}}}{\sqrt{2 \left(m^{(3-1.54)} - \frac{D_0 m^{1.5-(1.54)/2} \pi}{4\sqrt{A_0 K_0}} \right)}}$$

which for $m(T) = m_0(1 - [T/T_C]^{1.5})$ predicts an increase – rather than a decrease – of the skyrmion size with temperature and a divergence at some critical temperature [Tomasello *et al.*, Phys. Rev. B **97**, 060402(R) 2018]. So in conclusion, this simple theory based on analytical calculations and the typical scaling behavior of the magnetic parameters cannot explain our behavior. We would like to note that the scaling parameter could potentially differ for thin film systems with two-site anisotropy, such as

the case where the exponent $n \sim 2$ for K [H. Sato *et al.*, Phys. Rev. B **98**, 214428 (2018)., K. Alshammari *et al.*, Phys. Rev. B **104**, 224402 (2021).]. However, even in such cases, the qualitative conclusion remains unchanged.

Furthermore, our recent findings indicate that the interlayer stray field effect may play a critical role in stabilizing the topological spin textures in SyAFM systems [M. Bhukta *et al.*, arXiv:2303.14853]. As such, also this factor should be taken into consideration when modeling the size of these systems. However, it is important to emphasize that a comprehensive understanding of the underlying mechanism behind the drastic size modulation observed in SyAFM systems will require additional, extensive, and thorough experiments, which go beyond the scope of the current work. In particular, it will be necessary to independently measure the temperature dependence of various magnetic properties, which is experimentally challenging and requires work that has to be reserved for a future study. The objective of our work is to elucidate the effect of the gyrotropic force on diffusive motion. In light of this, we have added the following sentences with some references.

[page 11, lines 234-238 in the main text]

Fig. 5a can be related to the depinning energy density and the intrinsic diffusion coefficient, respectively. Whereas the effect explains well the systematic increase of the slope of the diffusion coefficient, the drastic size modification cannot be described easily by the existing theories^{36,49,50}. Assuming reasonable scaling factors of magnetic anisotropy^{51,52}, exchange stiffness^{53,54}, and interfacial DMI⁵²⁻⁵⁴, these theories predict that size increases at higher temperatures, which contradicts our experimental observation so that to explain this, one needs to study in a future work the temperature dependence of the individual parameters and their impact on the skyrmion size.

[page 11, lines 245-248 in the main text]

Another possible reason is the interlayer stray field effect which has often been neglected in the theory. Indeed, the recent experimental observation indicates that the interlayer dipole interaction could be a key enabler for stabilizing the topological spin structures⁵⁶ as Fig. 4b implies.

[Added References]

49. Rohart, S. & Thiaville, A. Skyrmion confinement in ultrathin film nanostructures in the presence of Dzyaloshinskii-Moriya interaction. Phys. Rev. B **88**, 184422 (2013).

50. Tomasello, R. *et al.* Origin of temperature and field dependence of magnetic skyrmion size in ultrathin nanodots. Phys. Rev. B **97**, 060402 (2018).

51. Sato, H. *et al.* Temperature-dependent properties of CoFeB/MgO thin films: Experiments versus simulations. Phys. Rev. B **98**, 214428 (2018).

52. Alshammari, K. *et al.* Scaling of Dzyaloshinskii-Moriya interaction with magnetization in

Pt/Co(Fe)B/Ir multilayers. Phys. Rev. B 104, 224402 (2021).

53. Rózsa, L., Atxitia, U. & Nowak, U. Temperature scaling of the Dzyaloshinsky-Moriya interaction in the spin wave spectrum. Phys. Rev. B 96, 094436 (2017).

54. Zhou, Y., Mansell, R., Valencia, S., Kronast, F. & van Dijken, S. Temperature dependence of the Dzyaloshinskii-Moriya interaction in ultrathin films. Phys. Rev. B 101, 054433 (2020).

56. Bhukta, M. *et al.* Homochiral antiferromagnetic merons, antimerons and bimerons realized in synthetic antiferromagnets. arXiv:2303.14853 [cond-mat] (2023).

Reviewer's comment

5. *5: English needs some attention. After addressing these issues, I can suggest its publication in Nature Comm.*

Our reply

We appreciate his/her careful reading. We thoroughly reviewed our manuscript including proof-reading it by a native speaker. We have revised typos and language deficiencies with tracking mode in the Word file, which is highlighted by the yellow marker. If the referee has any further suggestions on how to improve the language, we gladly would like to hear them so we can improve our work further.

Response to Reviewer #2

We are grateful to the reviewer for the critical and conscientious reading of our manuscript. In line with reviewer #1, the positive evaluation of our work that warrants publication in Nature Communications is highly appreciated. We have revised the manuscript according to the comments and suggestions. Our responses are listed below.

Reviewer's comment

In this work, the authors experimentally and theoretically study the thermally-activated diffusive motion of skyrmions in a synthetic antiferromagnetic system with low pinning. By tuning the compensation ratio of magnetic moments in the magnetic layers, they observe a more than 10 times larger diffusion coefficient for highly compensated antiferromagnetically-coupled skyrmions compared to the conventional ferromagnetic counterparts. The diffusive motion of skyrmions is a current hot topic in spintronics, as well as the results of this work may be potentially useful since future technological applications based on magnetic skyrmions require a full understanding of the skyrmion dynamics in the presence of thermal fluctuations. In general, this manuscript can be accepted after the following issues are clarified and corrected.

Our reply

We are pleased to learn that the reviewer has recognized the novelty and importance of our work and supported the publication of our manuscript in Nature Communications. We also thank the reviewer for summarizing the key points of our work.

Reviewer's comment

1. *Main issues:*

1) In Figure 2(d), one can see that the temperature dependence of the diffusion coefficient for the stacks lies in different temperature ranges. Is it possible to show the $D_{\text{diff}}-T$ relationship for the stacks with different compensation ratio in the same temperature range?

Our reply

In principle, the requested plots are feasible. Indeed, samples displaying a 75% compensation ratio and those with a 25% compensation ratio in Figure 2(a) of the main manuscript exhibit diffusive motion of magnetic skyrmions within an identical temperature range. Nevertheless, this specific temperature range is contingent solely upon the stabilization of magnetic skyrmions, accompanied by a size difference. Consequently, this implies a difference in the pinning energies inherent in magnetic

skyrmions, precluding a straightforward comparison. As a result, it is essential to account for the influence of size (as well as temperature), which corresponds to our Figure 5. In this figure, these effects (the difference in size and temperature) are corrected, allowing us to investigate how the intrinsic effect of gyrotropic force changes with respect to the compensation ratio.

Reviewer's comment

- 2. The authors claim that the measurable temperature range is limited due to constraints regarding the skyrmion nucleation, stability and the time resolution of their imaging setup. This may be right to a certain extent. However, I still would like the authors to argue more about this point or propose some potential methods to solve this problem, since the working window (i.e., the measurable temperature range) for each stack is very narrow, which is not beneficial for future applications.*

Our reply

We are grateful for this comment. Indeed in our current work, we have optimized the stack for low pinning combined with skyrmions that have a large enough size to be visible in Kerr microscopy in a temperature range accessible in our setup. The limited temperature range for stabilizing SyAFM skyrmions with a high compensation ratio can be attributed to the mechanism of stabilization being the stray field effect in our case. As the dipole interaction diminishes with an increasing compensation ratio, the temperature range permitting skyrmion stabilization becomes more restricted in comparison to ferromagnetic skyrmions. To address this challenge, one may consider employing DMI-stabilized skyrmions [F. Büttner *et al.*, *Sci. Rep.* **8**, 4464 (2018)], as they offer considerable thermal stability even at single-digit size scales. While a large DMI necessitates thin film crystallization, which could simultaneously lead to a rough surface and increased depinning energy from an experimental perspective, we believe that this issue can be resolved using the reverse sputtering technique. This method is commonly employed to achieve a clean surface, possibly allowing for the coexistence of a large DMI and minimal depinning. Consequently, a more extensive working window can be attained for SyAFM skyrmions with a high compensation ratio, enabling the enhanced diffusion to fully realize its potential. In order to emphasize this point, the above discussion is added to the main text as follows:

[page 14, lines 309-316 in the main text]

<From>

Even though we have judiciously used relatively large skyrmions that allow for the observation using optical Kerr-microscopy, our results of the enhancement of the diffusion coefficient will be key also for smaller skyrmions because of a smaller dissipative term $\alpha\Gamma$ indicating that the skyrmion is

close to the low damping regime shown Fig. 3b, which means being promising in terms of device scalability as well.

<To>

Whereas we have intentionally used relatively large skyrmions, that allow for the observation via optical Kerr-microscopy, the enhancement of the diffusion coefficient will be more significant for DMI-stabilized smaller skyrmions, owing to a smaller dissipative term $\alpha\Gamma$. Furthermore, the DMI-stabilized AFM skyrmions would maintain sufficient thermal stability even at single-digit size scales⁵⁵, which suggests the potential for a broader working temperature range in highly compensated states contrary to dipole-stabilized large skyrmions. Consequently, they offer promising implications for device scalability.

Reviewer's comment

- 3. In addition, I wonder to know why the measurable temperature range of the system with a compensation ratio of 60% is not between that of the systems with the compensation ratio of 75% and 25%, but is independently located on the far left.*

Our reply

As previously mentioned, the temperature range in which magnetic skyrmions exhibit diffusive motion is crucially dependent on their stabilization mechanism, which is governed by the effective anisotropy and dipole interaction. When controlling the magnetization compensation, changes occur not only in the volume and interlayer stray fields but also in the interfacial anisotropy due to modifications in the thickness ratio of $\text{Co}_{0.6}\text{Fe}_{0.2}\text{B}_{0.2}$ to $\text{Co}_{0.2}\text{Fe}_{0.6}\text{B}_{0.2}$, as well as the total thickness (as well as the dead layer). Consequently, the total magnetic energy contributing to the skyrmion stabilization exhibits non-monotonic and intricate behavior, making it challenging to discern an unambiguous trend in the temperature range.

Reviewer's comment

- 4. 2) Regarding the compensation dependence of the diffusion coefficient, the authors conduct a theoretical analysis based on Eqs. (2-4). To understand the steeper slope with increasing compensation ratio, in my opinion, finding the relationship between D_{dif}/T [or $\ln(D_{\text{dif}}/T)$] and m_{com} (or the mentioned ΔE) is the most intuitive and persuasive way. It is not straightforward to draw the conclusion that " ΔE increases monotonically with the compensation ratio" shown on page 10, line 207, from the current Eqs. 2 and 3. Therefore, can the authors provide an equation with the form of " $D_{\text{dif}}/T = \dots \Delta E$ "?*

Our reply

We appreciate this comment related to this potentially confusing point. It should be emphasized that the observation of a monotonically increasing ΔE is solely based on the experimental evidence presented in Fig. 2(b) (for the revised version), Fig. 4, and the previous observations [R. Gruber *et al.*, Skymion pinning energetics in thin film systems. Nat. Commun. **13**, 3144 (2022); J.-P. Tetienne *et al.*, Nanoscale imaging and control of domain-wall hopping with a nitrogen-vacancy center microscope. Science **344**, 1366 (2014).], and not derived from our Equations (2) and (3). To further elucidate this point, we have included the following sentence:

[page 10, lines 216-217 in the main text]

Note that Eq. (4) is solely based on Fig. 2(b), Fig. (4), and the previous observations^{47,48}.

Reviewer's comment

5. 3) *I understand that the authors using the collective domain motion to check the chirality of the spin texture in this system. I would like to know how to distinguish the skyrmion, skyrmion bubble and magnetic bubble in experiments.*

Our reply

We thank the referee for raising this important point. As established in previous studies, it is widely recognized that circular magnetic domains with trivial topology are unable to exhibit coherent movement by spin-orbit torques, and can only undergo vanishing or elongation [W. Jiang *et al.*, Science **349**, 283 (2015)]. Conversely, skyrmions and skyrmion bubbles are examples of non-trivial topological magnetic structures (Both are exactly the same objects in terms of topology). In our investigation, we have observed coherent motion of magnetic domains, as evidenced by the consistent ratio of white to black domains following multiple pulsing. Based on this observation, we can conclude that the magnetic spin structures in each of the layers of our samples exhibit non-trivial topological features, and therefore qualify as skyrmions.

Reviewer's comment

6. 4) *In the experimental section, the authors claim that the DMI is very weak in their system and the dipolar coupling effects play a key role in the stabilization of skyrmions. However, in the simulation, the dipole-dipole interaction is lacking in the spin Hamiltonian. The authors should clarify this point.*

Our reply

We thank the referee for highlighting this point. The skyrmions used in the experiments are of micrometer size, which is far away from what is accessible in atomistic spin model simulations. So instead of being able to do realistic simulations using atomistic spin models, we simulate small, DMI-stabilized skyrmions, and compare the dynamics qualitatively with experiments. This is an established approach to understanding qualitative behavior as done before for instance in [J. Zázvorka *et al.*, Nat. Nanotechnol. **14**, 658 (2019).]. For the small system sizes considered in the simulations, the dipole-dipole interactions have a negligible contribution, so they can be neglected. We have included the following sentence to make the point clearer:

[page 12, lines 254-255 in the main text]

In our stacks, the dipole interactions play a key role in the stabilization of skyrmions whereas their influence on the diffusion is negligible as shown in previous work²².

Reviewer's comment

7. *And also, can the authors provide one Figure or one Video to show the diffusive motion of SyAFM skyrmions in the simulation?*

Our reply

We have generated a video showing the diffusive motion which has been included as Supplementary Video 6 in the revised version. Also, we have added the following sentence.

[page 25, lines 570-571 in Methods]

An example of simulation for the SyAFM skyrmion diffusion is available in Supplementary Video 6.

Reviewer's comment

8. *Other issues:*

1) *On page 5, lines 102-104, there are many variables denoted by 'm' or 'M'. Please give the definition for each. Especially, do 'm_{CFB}' and 'm_{FCB}' indicate the saturation or unit magnetization of magnetic layers?*

Our reply

We have thought that the variables " m_{CFB} " and " m_{FCB} " can be omitted as they cannot be

determined independently in the experiment as well as to preclude any possible confusion due to the presence of many variables. Therefore, we have used only m_1 and m_2 as the saturation magnetic moment of the bottom and top FM layers. While the magnetic moment is the best indicator for the magnetic compensation ratio in the experiment (as the compensation is typically controlled by the thickness of the magnetic layer), the magnetization (the magnetic moments per volume) is the better indicator to control the compensation in the simulation owing to computational simplicity. In order to make this point clearer, we define those variables accordingly. The revised sentences are as follows:

[page 5, lines 103-105 in the main text]

<To>

Figure 1b shows magnetization curves (m - H curves) for the FM bi-layer and SyAFM systems, where the compensation ratio $m_{\text{Com}} = 1 - |m_1 + m_2| / (|m_1| + |m_2|)$, the saturation magnetic moment of the bottom FM layer m_1 , and that of the top FM layer m_2 can be determined³⁷.

[page 7, lines 147-148 in the main text]

<To>

The compensation of the synthetic antiferromagnet is varied by keeping magnetization (the magnetic moments per volume) M_2 fixed and varying M_1 between $-M_2$ and 0.

Reviewer's comment

9. 2) Please double check the main text and the supplementary information to eliminate typos and language deficiencies. For example, the last column of Table 2 should be the damping constant, denoted by alpha instead of A, and there is a wrong citation of Figures in Supplementary Note 3 [see "Based on the peak indicating the interlayer exchange coupling field in Fig. 2(b)"].

Our reply

We appreciate this comment. We thoroughly reviewed our manuscript including proof-reading by a native speaker. We have revised some typos and language deficiencies including the points raised here by the reviewer with tracking mode in the Word file, which is highlighted by the yellow marker. If the referee has further concrete suggestions to improve our wording, we are happy to hear about them and implement them to make our work even more accessible.

Reviewer's comment

10. 3) To improve the visibility of Figures and avoid any potential confusion, I suggest that the

authors can make some modifications to the Figures that include too many things, such as Figure 2 and Figure 5.

Our reply

Indeed, as also pointed out by reviewer #1 the figures need to be improved. In response in order to improve the accessibility of Fig. 2 and Fig. 5(a), we have strongly revised these figures. Specifically, with respect to Fig. 2, we have removed Fig. 2(a), (b), and (c), which were not necessary to focus on the findings of this work, as the same behavior has already been shown in previous works. Instead, we have integrated Fig. 2(a), which shows the skyrmions observed in our systems, into Fig. 1(a) to demonstrate that we have clearly observed the skyrmions. As for Fig. 5(a), we have adjusted the colors by lightening the error region and strengthening the color of the fitted line to make it more visible. We believe that these revisions have significantly improved the visibility of the figures compared to the previous versions. Please refer to the revised figures below:

<From>

<To>

[page 29, Figure 1 (a)]

<From>

<To>

<From>

<To>

<The summary of minor changes for the corresponding sentences in the main text>

[page 29, The caption of Fig. 1]

[page 30, The caption of Fig. 2]

[page 31, The caption of Fig. 5]

[page 6, lines 116-117 in the main text]

[page 6, line 126 in the main text]

[page 6, line 132 in the main text]

[page 6, line 135 in the main text]

Reviewer's comment

11. 4) Please make sure the labelling of variables is consistent. For instance, in the experimental section, the diffusion coefficient is denoted by ' $D_{\{dif\}}$ ', while in the theoretical part it is denoted by D/a^2 (see Figs. 2 and 3). The units of temperature used in experiment and theory are also different.

Our reply

We appreciate the reviewer carefully reading our work and indicating this confusing point. As shown below out, we have corrected the labeling as follows:

[page 31, Figure 3]

<From>

<To>

(a)

(b)

Response to Reviewer #3

We are grateful to the reviewer for the critical reading of our manuscript. In line with reviewers #1 and #2, the positive evaluation of our work that warrants publication in Nature Communications is highly appreciated. We have revised the manuscript according to the comments and suggestions. Our responses are listed below.

Reviewer's comment

Referee report on the manuscript entitled "Enhanced thermally-activated skyrmion diffusion with tunable effective gyrotropic force" by Takaaki Dohi et al. The topic of the manuscript is timely and of interest. Recently, it is clear that the conventional application of magnetic skyrmions such as memory are not competitive and then there is a great effort in looking for unconventional application where a skyrmion based technology can perform better. The work presented here is in context and it is a complete description of application of magnetic skyrmions for stochastic computing. Theory and experimental data well support the conclusions of the work. The idea of stochastic computing with skyrmions is not new, some of the authors were pioneering on this. Indeed, the stabilization of SyAF skyrmions is by itself an hot topic. I have enjoyed the reading and I'm positive with the publication. However, the following points should be addressed before.

Our reply

We are pleased to learn that the reviewer has appreciated the novelty and importance of our work and support the publication of our manuscript in Nature Communications. We also thank the reviewer for summarizing the key points of our work.

Reviewer's comment

- 1. The discussion "the SyAFM skyrmion shows a higher diffusion coefficient compared to its FM counterpart as shown in Fig. 3a. Hence, the behavior crucially depends on the magnetic parameters" is too generic. Here the key ingredients are the interlayer exchange coupling and the effective magnetizations.*

Our reply

We agree with the referee that, albeit being fundamentally correct, our statement is too generic and that it should be specified, which parameters are actually relevant here. Thus, we have reformulated the statement as follows:

[page 8, line 168-171 in the main text]

<From>

Hence, the behavior crucially depends on the magnetic parameters.

<To>

Hence, the behavior crucially depends on the interlayer exchange coupling (that causes both skyrmions to move as a single entity) and the respective saturation magnetizations (which determine the compensation of the layer magnetizations).

Reviewer's comment

- I suggest to add data on skyrmion motion driven by the current and add a comparison in the text. This can be used to identify the pinning-dominated diffusion regime.*

Our reply

Note that the possible method using current-induced skyrmion motion does not have good predictive power for whether the observed thermally-activated skyrmion diffusion is pinning-dominated or rather free as the dimension of the diffusion coefficient is $\text{m}^2 \text{s}^{-1}$ and the observed mean-squared displacement linearly scales with time as shown in Fig. 2(c). This means that the mean-squared velocity depends on time, and thus we should use the different scheme in order to check the motion regime. Hence, we first checked the temperature dependence of the diffusion coefficient. The diffusion coefficient for all the stacks exhibits a linear dependence on a semi-logarithmic scale (scale exponentially on the temperature) for the whole temperature range used in our experiments, which means that the observed thermally-activated skyrmion diffusion is in the pinning-dominated diffusion regime as the reviewer pointed out because the free diffusion should scale linearly on the temperature.

Reviewer's comment

- About the skyrmion size dependence on temperature the authors can refer to the papers: Rózsa, et al Phys. Rev. B 96, 094436, 2017 and Tomasello et al Phys. Rev. B 97, 060402(R) 2018 to verify the scaling of the DMI coefficient to explain the size effect. Considering that D is small, it is expected that also the scaling of other parameters can be important for the size calculation.*

Our reply

Revealing the critical factors determining the size of nearly compensated skyrmions is the important issues of the field of skyrmionics. Indeed, the reviewer #1 has raised the similar point. Here, we discuss

the contradiction between our experimental observations and the existing theories raised by the reviewer #3.

For a large, bubble-like skyrmion the radius follows this formula [S. Rohart and A. Thiaville, Phys. Rev. B **88**, 184422 (2013).]

$$R = \frac{\sqrt{\frac{A}{K}}}{\sqrt{2 \left(1 - \frac{D\pi}{4\sqrt{AK}} \right)}}$$

When we use the typical scaling behavior of the parameters [Rózsa, *et al.*, Phys. Rev. B **96** 094436, (2017).] $A = A_0 m^{1.54}$, $D = D_0 m^{1.54}$, and $K = K_0 m^3$ we get

$$R = \frac{\sqrt{\frac{A_0 m^{1.54}}{K_0 m^3}}}{\sqrt{2 \left(1 - \frac{D_0 m^{1.54} \pi}{4\sqrt{A_0 m^{1.54} K_0 m^3}} \right)}} = \frac{\sqrt{\frac{A_0}{K_0}}}{\sqrt{2 \left(m^{(3-1.54)} - \frac{D_0 m^{1.5-(1.54)/2\pi}}{4\sqrt{A_0 K_0}} \right)}}$$

which for $m(T) = m_0(1 - [T/T_C]^{1.5})$ predicts an increase – rather than a decrease – of the skyrmion size with temperature and a divergence at some critical temperature [Tomasello *et al.*, Phys. Rev. B **97**, 060402(R) 2018]. So in conclusion, this simple theory based on analytical calculations and the typical scaling behavior of the magnetic parameters cannot explain our behavior. We would like to note that the scaling parameter could potentially differ for thin film systems with two-site anisotropy, such as the case where the exponent $n \sim 2$ for K [H. Sato *et al.*, Phys. Rev. B **98**, 214428 (2018)., K. Alshammari *et al.*, Phys. Rev. B **104**, 224402 (2021).]. However, even in such cases, the qualitative conclusion remains unchanged.

Furthermore, our recent findings indicate that the interlayer stray field effect may play a critical role in stabilizing the topological spin textures in SyAFM systems [M. Bhukta *et al.*, arXiv:2303.14853]. As such, also this factor should be taken into consideration when modeling the size of these systems. However, it is important to emphasize that a comprehensive understanding of the underlying mechanism behind the drastic size modulation observed in SyAFM systems will require additional, extensive, and thorough experiments, which go beyond the scope of the current work. In particular, it will be necessary to independently measure the temperature dependence of various magnetic properties, which is experimentally challenging and requires work that has to be reserved for a future study. The objective of our work is to elucidate the effect of the gyrotropic force on diffusive motion. In light of this, we have added the following sentences and added the suggested references to the main text.

Figure 5a can be related to the depinning energy density and the intrinsic diffusion coefficient, respectively. Whereas the effect explains well the systematic increase of the slope of the diffusion coefficient, the drastic size modification cannot be described easily by the existing theories^{36,49,50}. Assuming reasonable scaling factors of magnetic anisotropy^{51,52}, exchange stiffness^{53,54}, and interfacial DMI⁵²⁻⁵⁴, these theories predict that size increases at higher temperatures, which contradicts our experimental observation so that to explain this, one needs to study in a future work the temperature dependence of the individual parameters and their impact on the skyrmion size.

[page 11, lines 245-248 in the main text]

Another possible reason is the interlayer stray field effect which has often been neglected in the theory. Indeed, the recent experimental observation indicates that the interlayer dipole interaction could be a key enabler for stabilizing the topological spin structures⁵⁶ as Fig. 4b implies.

[Added References]

49. Rohart, S. & Thiaville, A. Skyrmion confinement in ultrathin film nanostructures in the presence of Dzyaloshinskii-Moriya interaction. *Phys. Rev. B* 88, 184422 (2013).

50. Tomasello, R. *et al.* Origin of temperature and field dependence of magnetic skyrmion size in ultrathin nanodots. *Phys. Rev. B* 97, 060402 (2018).

51. Sato, H. *et al.* Temperature-dependent properties of CoFeB/MgO thin films: Experiments versus simulations. *Phys. Rev. B* 98, 214428 (2018).

52. Alshammari, K. *et al.* Scaling of Dzyaloshinskii-Moriya interaction with magnetization in Pt/Co(Fe)B/Ir multilayers. *Phys. Rev. B* 104, 224402 (2021).

53. Rózsa, L., Atxitia, U. & Nowak, U. Temperature scaling of the Dzyaloshinsky-Moriya interaction in the spin wave spectrum. *Phys. Rev. B* 96, 094436 (2017).

54. Zhou, Y., Mansell, R., Valencia, S., Kronast, F. & van Dijken, S. Temperature dependence of the Dzyaloshinskii-Moriya interaction in ultrathin films. *Phys. Rev. B* 101, 054433 (2020).

56. Bhukta, M. *et al.* Homochiral antiferromagnetic merons, antimerons and bimerons realized in synthetic antiferromagnets. arXiv:2303.14853 [cond-mat] (2023).

Reviewer's comment

4. *The data on BLS for evaluating the DMI should be included. Please include also the parameters used for the estimation.*

Our reply

We agree with this comment. Showing data for the Brillouin light scattering (BLS) data needs a

space in the manuscript. Hence, we add the BLS data as well as the parameters used for evaluating the DMI in the additional Supplementary Note 6.

Reviewer's comment

5. *The discussion about the chirality should be improved. It is not completely clear to me. I suggest adding a 2d phase diagram of the chirality considering interlayer exchange and composition.*

Our reply

We also agree with this comment. The ambiguity should be avoidable by the detailed discussion. In order to investigate the domain wall chirality transition, we have run the micromagnetic simulation using MuMax3, and the below detailed discussion is put in Supplementary Note 7.

<Supplementary Note 7: Chirality evaluation for bi-layer systems >

In order to evaluate the chirality of our SyAFM bi-layer systems, we use micromagnetic simulations with the software MuMax¹⁷. We show the parameter list used for the simulation in Supplementary Table 1, corresponding to the experimental value for the SyAFM system with 90% magnetic compensation. Note that the interlayer magnetic dipole interaction (stray field effect between the top FM and the bottom FM layers) to compete with the interfacial DMI and the interlayer exchange coupling would play a key role in determining the chirality of the system in the case of a relatively small DMI. Thus, we employ the custom fields functionality to incorporate the interlayer exchange coupling field, allowing us to separate the top and bottom FM layers with a finite non-magnetic interlayer¹⁸. As a consequence, the interlayer magnetic dipole interaction is successfully taken into account in our simulation.

To reveal the effect of interlayer exchange coupling and magnetic compensation on the domain wall (DW) chirality, we make a two-dimensional (2D) phase diagram of the DW angle by varying these parameters. The built-in function “two domain” is used to initialize the domain state for antiparallel configuration between the top and bottom FM layers with the DW at the center. Then, we also use the built-in function “relax” to obtain an equilibrium DW angle.

Supplementary Fig. 6 (a), (b), and (c) show the DW angle, φ for the up-down DW of the top FM layer, for the down-up DW of the bottom FM layer, and the corresponding domain configuration, respectively. The φ is defined as an azimuthal angle with regard to the x -axis shown in Fig. 6 (c). In supplementary Fig. 6 (a) and (b), the white circles present the experimental parameter for the stacks used in this work. Concerning the J_{int} dependence, we find the stable AFM coupling DW for the J_{int} roughly larger than $5 \mu\text{J m}^{-2}$. The φ changes with increasing J_{int} and saturates above some threshold

J_{int} depending on the magnetic compensation, m_{Com} . As the experimentally obtained J_{int} is unambiguously larger than the threshold J_{int} , hereafter, we focus on the compensation ratio dependence of φ . Based on the simulation, we can divide the phase into four regions regarding chirality. 1: Left-handed Néel for m_{Com} larger than 90%, 2: Bloch-Néel (left-handed) hybrid for m_{Com} between 90 and 60%, 3: Right-handed Néel for m_{Com} less than 60%, 4. Hybrid Néel for the FM bi-layer system as shown in supplementary Fig. 6(c). The interfacial DMI determines the chirality for the highly compensated state simply because both the volume dipole and interlayer dipole interactions are small, even if the DMI magnitude is relatively small. The obtained left-handed chirality for the highly compensated state is consistent with our current-induced collective DW motion. Decreasing m_{Com} gradually increases the contribution of both dipole interactions; thus, the DW chirality gradually changes. With further decreasing m_{Com} , intriguingly, the DW chirality is determined by the interlayer dipole interaction due to the flux closure for domain-domain wall interaction¹⁹ by overcoming the volume dipole contribution, which is a unique feature for the bi-layer system. The chirality is right-handedness in our case (as M_s for the top layer is larger than the bottom counterpart). On top of this, the AFM coupling between DWs induces coherent right-handed chirality for the SyAFM systems. In contrast, the hybrid Néel DW is stabilized for the FM bi-layer system as previously demonstrated¹⁹. In summary, we demonstrate that the Néel DW is stable in the stacks except for the $m_{\text{Com}} = 75\%$ case, which would have the Bloch-Néel hybrid DW chirality.

Supplementary Table 1 | Micromagnetic simulation parameter list

World size (top, interlayer, bottom thickness)	512x256x3 nm ³ (1 nm, 1 nm, 1 nm)
Cell size	2x2x1 nm ³
Periodic boundary condition	32x32x0
Spontaneous magnetization, M_s (top, bottom)	(1.0+x, 1.0-x) T
Effective perpendicular anisotropy field, $\mu_0 H_K^{\text{eff}}$	45 mT
Interfacial DMI, D_i	-0.05 mJ m ⁻²
Exchange stiffness, A_s	6 pJ m ⁻¹
Interlayer exchange coupling energy J_{int}	x mJ m ⁻²
Gilbert damping, α	1
External perpendicular magnetic field $\mu_0 H_z$	0.1 mT
Initialization	two domain top: (0, 0, 1, 1, 1, 0, 0, 0.5, -1) bottom: (0, 0, -1, 1, 1, 0, 0, 0.5, 1)

Supplementary Figure 6 | The 2D phase diagram of the domain wall (DW) angle φ defined as an azimuthal angle on the basis of the x -axis. **a**, Up-down DW for a top FM layer. The third and fourth quadrants are used. Red, green, and blue colors correspond to -180 , -90 , and 0 degrees, respectively. **b**, Down-up DW for a bottom FM layer. The first and second quadrants are used. Red, green, and blue colors correspond to 180 , 90 , and 0 degrees, respectively. **c**, Domain and DW configuration for various values of the magnetic compensation.

<Added Supplementary References>

17. Vansteenkiste, A. *et al.* The design and verification of MuMax3. *AIP Advances* **4**, 107133 (2014).
18. Leliaert, J., Mulkers, J., Gypens, P. & Van Waeyenberge, B. MUMAX3-WORKSHOP SESSION 4, 55-64. <https://mumax.ugent.be/mumax3-workshop/> (2020).
19. Hrabec, A. *et al.* Current-induced skyrmion generation and dynamics in symmetric bilayers. *Nat. Commun.* **8**, 15765 (2017).

Note that the topology defined as the Néel vector is exactly the same for all the stacks, which means that it is necessary to investigate the current-induced motion to validate if the spin structures are topologically non-trivial or not. As we mentioned in the responses for the reviewer #2, topologically trivial magnetic domains cannot move coherently by the spin-orbit torque (the topologically trivial domains either vanish or elongate) [W. Jiang *et al.*, *Science* **349**, 283 (2015)]. Therefore, the observed coherent magnetic domain motion in our samples indicates that they are not topologically trivial. In order to emphasize this point, we have revised the sentences as follows:

[page 11, lines 248-253 in the main text]

Note that the topology of the skyrmions is fixed to be 1 due to the antiferromagnetic coupling domain walls forming flux closure⁵⁷, which is corroborated by the current-induced coherent and isotropic collective motion of domains¹⁴ (see Methods, Supplementary Video 4 and 5). Nevertheless, the chirality hinges on the compensation ratio owing to competing energies for the interfacial DMI, the interlayer exchange coupling, and the interlayer stray field (see Supplementary Note 7).

[Added Reference]

57. Meijer, M. J. *et al.* Magnetic Chirality Controlled by the Interlayer Exchange Interaction. *Phys. Rev. Lett.* **124**, 207203 (2020).

Reviewer's comment

6. *It is not clear which is the peak the authors are referring in Supplementary Fig. 1 (b).*

Our reply

We appreciate the reviewer's suggestion of the confusing point for supplementary Fig. 1(b). What we mention as the peak corresponds to the interlayer exchange coupling field, $\mu_0 H_{\text{int}}$, defined as the switching field which induces the ferromagnetic-antiferromagnetic transition. To make things clear, we have put an additional figure on the right side of supplementary Fig. 1(b). The revised figure is as follows:

[page 4, Supplementary Fig. 1]

<From>

<To>

Accordingly, we have revised the caption and the related sentence in our revised Supplementary Information as follow:

[page 4, The caption of Supplementary Fig. 1]

<From>

Supplementary Figure 1 | Ir thickness dependence of Kerr hysteresis curve. **a**, Stack structure used for investigating the Ir thickness dependence of the interlayer exchange coupling. The unit of the number shown in parenthesis is nanometer (nm). **b**, Ir thickness dependence of the Kerr hysteresis loops.

<To>

Supplementary Figure 1 | Ir thickness dependence of Kerr hysteresis curve. **a**, Stack structure used for investigating the Ir thickness dependence of the interlayer exchange coupling. The unit of the number shown in parenthesis is nanometer (nm). **b**, Ir thickness dependence of the Kerr hysteresis loops and the oscillation of interlayer exchange coupling field $\mu_0 H_{int}$.

[page 3, Note 1]

<From>

As shown in supplementary Fig. 1(b), the stack with $t_{\text{Ir}} = 1.2$ nm shows the second peak, which is consistent with previous work¹.

<To>

As shown in supplementary Fig. 1(b), the stack with $t_{\text{Ir}} = 1.2$ nm shows the second peak of the interlayer exchange coupling field $\mu_0 H_{\text{int}}$ defined by a magnetic field that induces the spin flip-like antiferromagnetic-ferromagnetic transition, which is consistent with previous work¹.

Reviewers' Comments:

Reviewer #1:

Remarks to the Author:

Authors made substantial efforts in addressing my comments and the comments from the other two reviewers. In my opinion, the quality of this revised manuscript has been substantially improved and meets the publishing criteria of Nature Communications.

Reviewer #2:

Remarks to the Author:

The authors have provided in their Response to Referees Letter convincing arguments for the acceptance of the manuscript in Nature Communications. I would suggest, however, that the authors carefully double check the Figure 3, including the labels and figure captions. We can see that the temperature in Fig.3 is in the range of $[0, 1.2]$ K, which may not be consistent with the experiments. In fact, the color code is different from that in Fig.2. The authors should update the figure caption on page 32, line 674, accordingly. Regarding the title of the y-axis, could the authors please give the definition of the variable "a"? In addition, there is no Fig.2d in the current version, so the authors should update the description on page 9, line 185.

In summary, I recommend the publication of the revised manuscript in Nature Communications after the authors confirm the above issues.

Reviewer #3:

Remarks to the Author:

The authors addressed my previous concerns and I recommend the manuscript for publication.

Response to Reviewer #1

We are grateful to the reviewer for the critical reading of our manuscript. In line with reviewers #2 and #3, the positive evaluation of our work that warrants publication in Nature Communications is highly appreciated. Our responses are listed below.

Reviewer's comment

Authors made substantial efforts in addressing my comments and the comments from the other two reviewers. In my opinion, the quality of this revised manuscript has been substantially improved and meets the publishing criteria of Nature Communications.

Our reply

We are very pleased to learn that our revisions have resolved the reviewer's concerns and the reviewer now supports the publication of our manuscript in Nature Communications.

Response to Reviewer #2

We are grateful to the reviewer for the critical and conscientious reading of our manuscript. In line with reviewers #1 and #3, the positive evaluation of our work that warrants publication in Nature Communications is highly appreciated. We have revised the manuscript according to the comments and suggestions. Our responses are listed below.

Reviewer's comment

The authors have provided in their Response to Referees Letter convincing arguments for the acceptance of the manuscript in Nature Communications.

Our reply

We are very pleased to learn that our revisions have resolved the reviewer's concerns and the reviewer now supports the publication of our manuscript in Nature Communications.

Reviewer's comment

- 1. I would suggest, however, that the authors carefully double check the Figure 3, including the labels and figure captions. We can see that the temperature in Fig.3 is in the range of $[0, 1.2]$ K, which may not be consistent with the experiments. In fact, the color code is different from that in Fig.2. The authors should update the figure caption on page 32, line 674, accordingly. Regarding the title of the y-axis, could the authors please give the definition of the variable "a"? In addition, there is no Fig.2d in the current version, so the authors should update the description on page 9, line 185.*

Our reply

We are grateful for the conscientious reading and the suggestions. The temperature range is indeed smaller than the experimental values. However, it can be easily scaled by changing all relevant energies (exchange stiffness, magnetic anisotropies, and Dzyaloshinskii-Moriya interaction) such that the skyrmions are stable at room temperature. The point is, that while energetics do not change if we scale all relevant energy scales including temperature, their diffusive motion becomes much larger. Therefore, we chose rather small values overall, in order to make the simulation computationally feasible.

Regarding the other concerns; the color code (to make all the figures' color code consistent), the variable "a", and the description relevant to Fig. 3, we have revised them based on the suggestions as

follows. In particular, for the color code, all the figures now have a consistent color code regarding the magnetic compensation with the revised Fig. 3 (, which is also consistent with that of Fig.2).

[page 31, Figure 3]

<To>

[page 9, line 185 in the main text]

<From>

...in Fig. 2d cannot be described within this simple model. Here we need to take pinning...

<To>

...in Fig. 2b cannot be described within this simple model. Here we need to take pinning...

[page 25, lines 563-564 in METHODS]

<From>

The saturation magnetic moments in the two monolayers are given by $m_1 = M_1 a^2$ and $m_2 = M_2 a^2$.

<To>

The saturation magnetic moments in the two monolayers are given by $m_1 = M_1 a^2$ and $m_2 = M_2 a^2$, where a is the lattice constant.

[page 31, lines 672-673 in the main text]

<From>

Atomistic simulations within a model using computationally feasible small skyrmions. (a) The temperature dependence of the diffusion coefficient.

<To>

Atomistic simulations within a model using computationally feasible small skyrmions, where a

denotes the lattice constant²⁹. (a) The temperature dependence of the diffusion coefficient.

Reviewer's comment

2. *In summary, I recommend the publication of the revised manuscript in Nature Communications after the authors confirm the above issues.*

Our reply

We appreciate further suggestions to improve the quality of our manuscript. We believe that the present revisions improve the quality of our manuscript and thus our manuscript is now suitable for publication in Nature Communications.

Response to Reviewer #3

We are grateful to the reviewer for the critical reading of our manuscript. In line with reviewers #1 and #2, the positive evaluation of our work that warrants publication in Nature Communications is highly appreciated. Our responses are listed below.

Reviewer's comment

The authors addressed my previous concerns and I recommend the manuscript for publication.

Our reply

We are very pleased to learn that our revisions have resolved the reviewer's concerns and the reviewer now supports the publication of our manuscript in Nature Communications